# Evidence for dopaminergic involvement in endogenous modulation of pain relief

Simon Desch[1,2]*, Petra Schweinhardt[3], Ben Seymour[4], Herta Flor[1], Susanne Becker[1,2,3]*

[1]Institute of Cognitive and Clinical Neuroscience, Central Institute of Mental Health, Medical Faculty Mannheim, Heidelberg University, Mannheim, Germany; [2]Clinical Psychology, Department of Experimental Psychology, Heinrich Heine University Düsseldorf, Düsseldorf, Germany; [3]Integrative Spinal Research, Department of Chiropractic Medicine, Balgrist University Hospital, University of Zurich, Zurich, Switzerland; [4]Wellcome Centre for Integrative Neuroimaging, John Radcliffe Hospital, Oxford, United Kingdom

**Abstract** Relief of ongoing pain is a potent motivator of behavior, directing actions to escape from or reduce potentially harmful stimuli. Whereas endogenous modulation of pain events is well characterized, relatively little is known about the modulation of pain relief and its corresponding neurochemical basis. Here, we studied pain modulation during a probabilistic relief-seeking task (a 'wheel of fortune' gambling task), in which people actively or passively received reduction of a tonic thermal pain stimulus. We found that relief perception was enhanced by active decisions and unpredictability, and greater in high novelty-seeking trait individuals, consistent with a model in which relief is tuned by its informational content. We then probed the roles of dopaminergic and opioidergic signaling, both of which are implicated in relief processing, by embedding the task in a double-blinded cross-over design with administration of the dopamine precursor levodopa and the opioid receptor antagonist naltrexone. We found that levodopa enhanced each of these information-specific aspects of relief modulation but no significant effects of the opioidergic manipulation. These results show that dopaminergic signaling has a key role in modulating the perception of pain relief to optimize motivation and behavior.

**\*For correspondence:**
simon.desch@zi-mannheim.de (SD);
sbecker@uni-duesseldorf.de (SB)

**Competing interest:** The authors declare that no competing interests exist.

## Editor's evaluation

This is an important paper that is of interest to researchers interested in the psychological and neurochemical mechanisms of pain and pain relief. It shows that the perception of pain relief is modulated by controllability, surprise and novelty seeking. Moreover, these modulations are influenced by dopaminergic but not by opioidergic manipulations. These findings are supported by convincing evidence.

## Introduction

One of the most powerful and universally appreciated aspects of being in pain is the desire for relief, and the positive sensation of pleasure once achieved. However, in contrast to pain itself, much less is known about how the perception of relief is modulated by various behavioral and motivational factors (*Becker et al., 2015*; *Leknes et al., 2008*). Theories of pain have suggested that one reason pain itself is modulated is to help optimize the way in which it directs protective behavior (*Walters and Williams, 2019*). For instance, if pain is increased when learning and responding to it, but reduced in

situations in which pain might actually interfere with optimal behavior will have a greater long-term benefit (*Fields, 2018*; *Seymour, 2019*). Whether this principle extends to relief has not been tested.

Endogenous modulation of pain involves a number of different processes mediated by distinct descending signaling pathways, and involves at least two critical neurochemical systems: opioidergic and dopaminergic (*Bannister, 2019*). For instance, opioid signaling has been shown to play a key role in behavioral relief motivation (in rodents, *Navratilova et al., 2015b*), placebo analgesia (*Benedetti, 1996*; *Eippert et al., 2009*), conditioned pain modulation (*King et al., 2013*), and relief perception (*Sirucek et al., 2021*). Dopaminergic signaling has clearly been shown to play a role in conditioned place preference induced by pain relief (through activity in midbrain dopaminergic neurons, *Navratilova et al., 2012*; *Navratilova et al., 2015a*; *Xie et al., 2014*), suggesting similar mechanisms as in the well-studied role of dopamine in food rewards (dopaminergic 'wanting' versus opioidergic 'liking'; *Barbano and Cador, 2006*; *Barbano and Cador, 2007*; *Berridge et al., 2009*; *Smith et al., 2011*). Dopaminergic signaling is also implicated in inhibition of pain by extrinsic rewards, that is rewards external to the pain system (*Becker et al., 2013*). Hence, whilst it is clear that both opioidergic and dopaminergic signaling play core roles in relief motivation, it isn't known which system primarily shapes perceived relief as a function of motivation.

The aim of the present study was therefore first to better characterize information processing aspects of relief motivation, and second to investigate the roles of dopaminergic and opioidergic signaling in pain relief perception and modulation. We expected that pain relief would be modulated by the value of information it carries, as hence enhanced by (i) active vs passive reception and thus controllability, since this reflects potential to exploit relief information; (ii) unpredictability, since this reflects the extra information carried by surprising events, and (iii) trait novelty-seeking, since this reflects individual information sensitivity. At the same time, we aimed to identify the potential role of dopamine and opioids for each of these factors, in particular to explore whether increased dopamine availability would enhance endogenous pain relief under these conditions, and whether modulation could be reduced by blocking opioid receptors. We expected increased dopamine availability to enhance phasic release of dopamine in response to rewards, and hence, to increase the effect of active compared to passive reception of pain relief. In contrast, we expected the inhibition of endogenous opioid signaling to decrease the effect of active controllability on pain relief. The latter is based on the observation that blocking of opioid receptors attenuates other types of endogenous pain inhibition such as placebo analgesia (*Benedetti, 1996*; *Eippert et al., 2009*) or conditioned pain modulation (*King et al., 2013*). Finally, we aimed to identify whether an increase or decrease in the modulation of perceived relief by dopamine and opioids is reflected in corresponding increases or decreases in the selection of a more advantageous option, that is decision-making during probabilistic learning.

To test these hypotheses, we employed a previously developed wheel of fortune task utilizing relief of a tonic capsaicin-sensitive thermal pain stimulus as 'wins', and allowing to quantify endogenous pain inhibition induced by gaining pain relief in active versus passive conditions (*Becker et al., 2015*). To test the roles of dopamine and opioids, we analyzed and report data of N=28 healthy volunteers who ingested either a single dose of the dopamine precursor levodopa (150 mg), the opioid antagonist naltrexone (50 mg), or placebo in separate testing sessions (double-blinded, placebo controlled cross-over, i.e. within-subjects, design). To allow also the assessment of reinforcement learning, a probabilistic reward schedule associated with the participants' choices in the wheel of fortune was implemented.

## Results

### Endogenous modulation of active pain relief seeking under placebo

To test whether playing the wheel of fortune game induced endogenous pain inhibition by gaining pain relief during active (controllable) decision-making, a test condition in which participants actively engaged in the game and 'won' relief of a tonic thermal pain stimulus in the game was compared to a control condition with passive receipt of the same outcomes (*Figure 1*). As a further comparator the game included an opposite condition in which participants received *increases* of the thermal stimulation as punishment. This active loss condition was also matched by a passive condition involving receipt of the same course of nociceptive input. We implemented two outcome measures, an explicit rating of perceived intensity after pain relief or increase, and a behavioral measure of perceptual

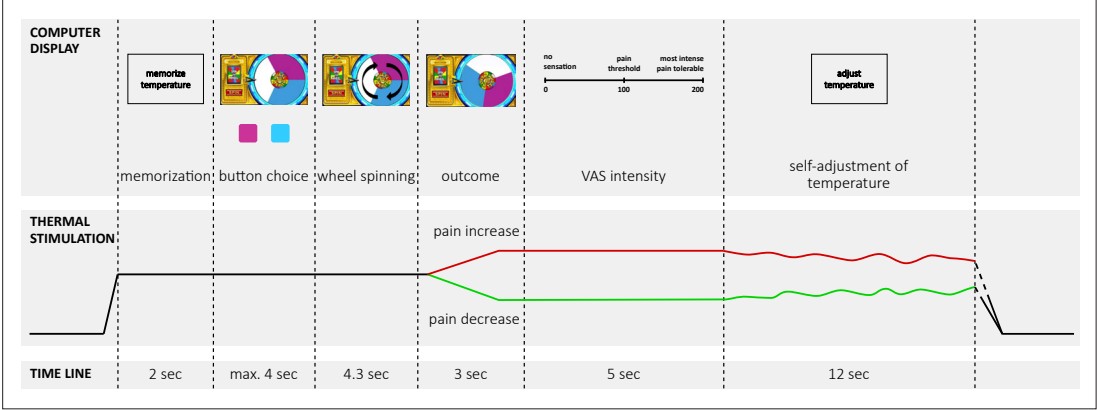

**Figure 1.** Time line of one trial with active decision-making (test trials) of the wheel of fortune game. Experimental pain was implemented using contact heat stimulation on capsaicin sensitized skin on the forearm. In each trial, the temperature increased from a baseline of 30 °C to a predetermined moderately painful stimulation intensity perceived as moderately painful. In each testing session, one of the two colors (pink and blue) of the wheel was associated with a higher chance to win pain relief (counterbalanced across subjects and drug conditions). Pain relief (win) as outcome of the wheel of fortune game (depicted in green) and, pain increase (loss; depicted in red) were implemented as phasic changes in stimulation intensity offsetting from the tonic painful stimulation. Based on a probabilistic reward schedule for these outcomes, participants could learn which color was associated with a better chance to win pain relief. In passive control trials and neutral trials participants did not play the game but had to press a black button after which the wheel started spinning and landed on a random position with no pointer on the wheel. Trials with active decision-making were matched by passive control trials without decision making but the same nociceptive input (control trials), resulting in the same number of pain increase and pain decrease trials as in the active condition. In neutral trials the temperature did not change during the outcome interval of the wheel. Two outcome measures were implemented in all trial types: (i) after the phasic changes during the outcome phase participants rated the perceived momentary intensity of the stimulation on a visual analogue scale ('VAS intensity'); (ii) after this rating, participants had to adjust the temperature to match the sensation they had memorized at the beginning of the trial, i.e. the initial perception of the tonic stimulation intensity ('self-adjustment of temperature'). This perceptual discrimination task served as a behavioral assessment of pain sensitization and habituation across the course of one trial. One trial lasted approximately 30 s, phasic offsets occurred after approximately 10 s of tonic pain stimulation. Adapted from *Becker et al., 2015*. Figure 1 is reproduced from Figure 1 in *Becker et al., 2015*.

sensitization or habituation to the underlying tonic stimulation within each trial of the game (see *Figure 1*). The effects of controllability on pain perception were tested in separate linear mixed effects models predicting the outcome measures by the outcome condition, the trial type (active test trials vs. passive control trials), and their interaction in each drug condition. Comparing the effects of active versus passive trials between the pain relief and the pain increase condition (interaction 'outcome × trial type') allowed also to test for unspecific effects of arousal and/or distraction: if the effects seen in the active compared to the passive condition were due to such unspecific effects then actively engaging in the game should equally affect pain in both, win and lose trials. In contrast, if the effects were due to increased controllability then we expected to see pain inhibition in win trials but equal or increased pain perception in lose trials. We used post-hoc comparisons to test direction and significance of differences in either outcome condition and report standardized effect sizes (*d*) for these differences. Note that all reported effect sizes account for random variation within the sample, providing an estimate for the underlying population; due to considerable variance between participants in the present study, this resulted in comparatively small effect sizes.

## Ratings of perceived pain

Replicating previous results, in the placebo (i.e. non-drug) condition participants rated the thermal stimulation as less intense after actively winning pain relief compared to the passive control condition, as rated on visual analogue scales (VAS) from 'no sensation' (0) over 'just painful' (100) to 'most intense pain tolerable' (200). Furthermore, participants also rated the stimulation as more intense after actively losing compared to the passive control condition (*Figure 2A*; interaction 'outcome × trial type', $F_{(1,1040)}=64.14$, p<0.001; pairwise comparisons: win: test vs. control p<0.001, standardized effect size *d*=0.16; lose: test vs. control, p<0.001, *d*=0.27). This shows that perception of both relief and pain are enhanced by active (instrumental) controllability, as hypothesized.

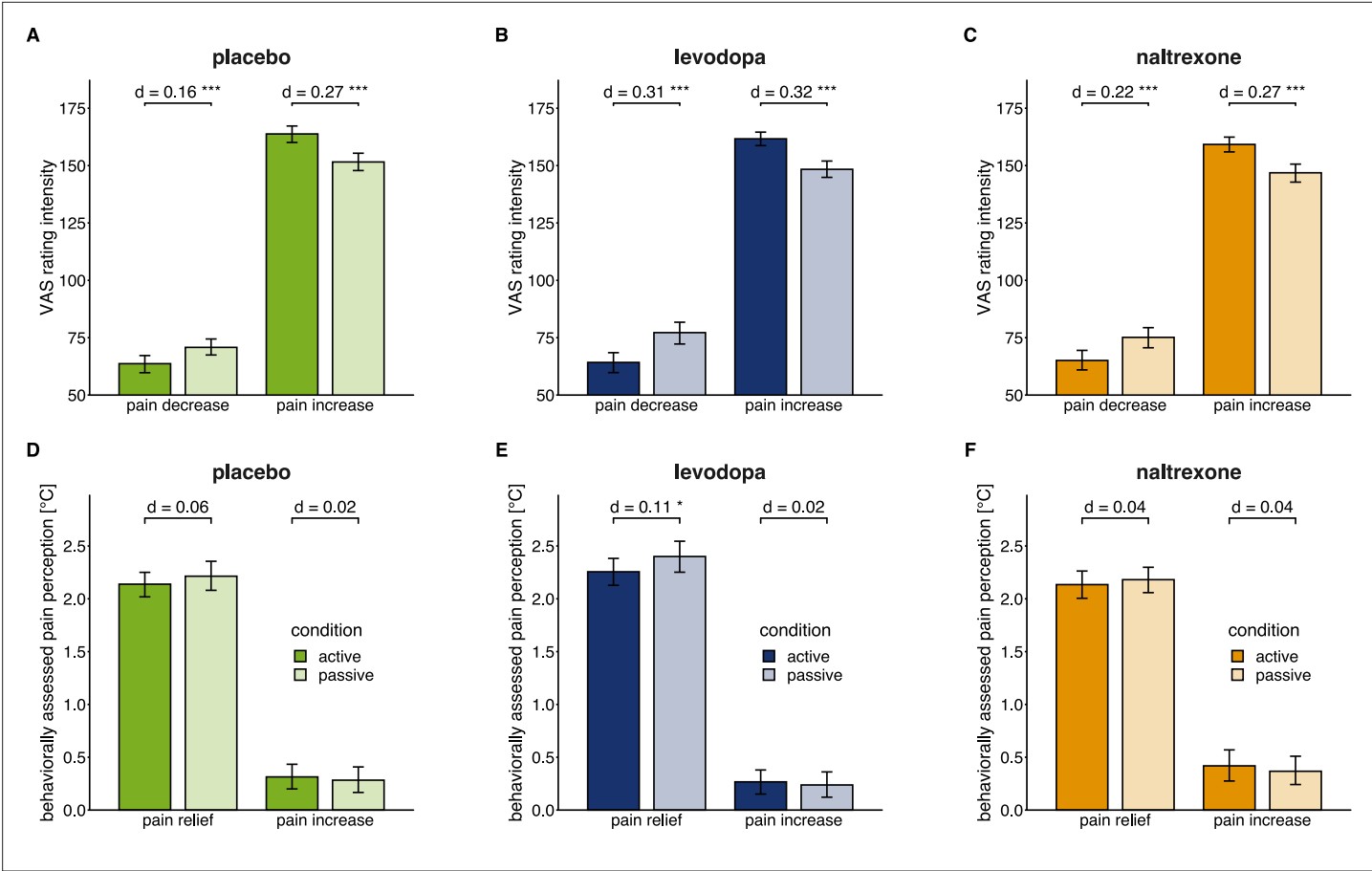

**Figure 2.** Effects of active versus passive condition after pain relief and pain increase in each drug condition. Means (bars) and 95% confidence intervals of means (error bars) for VAS pain intensity ratings (**A, B, C**) and behaviorally assessed pain perception (**D, E, F**; within-trial sensitization in pain perception in °C) for each drug session (placebo: n=28, levodopa: n=27, naltrexone: n=28). d indicates the standardized effect size after controlling for random effects and residual variance. ** p<0.01, *** p<0.001, for post-hoc comparisons of test versus control trials.

The online version of this article includes the following figure supplement(s) for figure 2:

**Figure supplement 1.** Individual level effects of active versus passive condition after pain relief and pain increase in each drug condition.

## Behaviorally assessed pain perception

In addition to the VAS ratings, participants performed a validated perceptual task (*Becker et al., 2011*; *Kleinböhl et al., 1999*) allowing to assess perception of the underlying tonic pain stimulus, which is specifically sensitive to perceptual sensitization and habituation. In this procedure, participants re-adjust the stimulation temperature themselves after the outcome of the wheel of fortune to match their perception at the beginning of each trial. Positive values (i.e. lower self-adjusted temperatures compared to the stimulation intensity at the beginning of the trial) indicate perceptual sensitization across the course of one trial of the game, negative values indicate habituation. For tonic stimulation at intensities that are perceived as painful, perceptual sensitization is expected to occur (*Kleinböhl et al., 1999*). Differences between the outcome conditions (win, lose) reflect the effect of the phasic changes on the perception of the underlying tonic stimulus. Differences between active and passive trials reflect the effect of controllability on the perception within each outcome condition. In contrast to the VAS ratings, behaviorally assessed pain perception did not significantly differ between test and control trials after winning as well as after losing in the placebo condition (*Figure 2D*; interaction 'outcome × trial type', $F_{(1, 1040)}=2.53$, p=0.112).

# Levodopa increases endogenous pain modulation by active relief with no significant effects of naltrexone on the modulation

We next examined whether endogenous modulation of pain perception within the wheel of fortune game was affected by a levodopa and naltrexone.

## Manipulation check: successful blinding of drug conditions

After the intake of *levodopa,* one participant reported a weak feeling of nausea and headaches at the end of the experimental session. In 32 out of 83 experimental sessions subjects reported tiredness at the end of the session. However, the frequency did not significantly differ between the three drug conditions ($\chi^2$ (2)=2.17, p=0.337) or between the placebo condition compared to the levodopa and naltrexone condition ($\chi^2$ (1)=1.06, p=0.304). No other side effects were reported. To ensure that participants were kept blinded throughout the testing, they were asked to report at the end of each testing session whether they thought they received levodopa, naltrexone, placebo, or did not know. In 43 out of 83 sessions that were included in the analysis (52%), participants reported that they did not know which drug they received. In 12 out of 28 sessions (43%), participants were correct in assuming that they had ingested the placebo, in 6 out of 27 sessions (22%) levodopa, and in 2 out of 28 sessions (7%) naltrexone. The amount of correct assumptions differed between the drug conditions ($\chi^2$ (2)=7.70, p=0.021). However, post-hoc tests revealed that neither in the levodopa nor in the naltrexone condition participants guessed the correct pharmacological manipulation significantly above chance level (p's>0.997) and the amount of correct assumptions did not differ significantly between placebo compared to levodopa and naltrexone sessions on the other hand ($\chi^2$ (1)=0.11, p=0.737), suggesting that the blinding was successful.

## Effects of pharmacological manipulations on endogenous modulation by active controllability

We next examined whether endogenous modulation of pain perception within the wheel of fortune game was affected by a levodopa and naltrexone. In addition to the models testing effects of controllability on pain perception within each drug condition, we calculated pain modulation in test trials for both outcome measures as the difference of each test trial to the mean of control trials for the respective outcome condition for each participant. We fitted linear mixed effects models to predict pain modulation by drug condition, outcome, and their interaction. As an additional covariate of no interest we included session order (and respective interaction effects) in these models, as the temporal order of sessions (independent of the order of the application of the drugs) was found to have a differential effect on win and lose outcomes (see *Figure 3—figure supplement 1*).

## Ratings of perceived pain

As in the placebo condition, participants rated the thermal stimulation as significantly less intense after active relief winning in the wheel of fortune task, and as significantly more intense after receiving phasic pain increases ('losing') compared to the respective passive control condition under levodopa (pairwise comparisons: win: test vs. control p<0.001, *d*=0.31; lose: test vs. control, p<0.001, *d*=0.32; *Figure 2B*) as well as naltrexone (pairwise comparisons: win: test vs. control p<0.001, *d*=0.22; lose: test vs. control, p<0.001, *d*=0.27; *Figure 2C*).

Moreover, the effect of active relief or increases on pain modulation was differentially modulated by the drugs (*Table 1*; interaction 'drug × outcome', *F*(2, 1587.30)=4.52, p=0.011). Specifically, the

**Table 1.** Means and standard deviation of means for pain modulation in VAS ratings of perceived intensity and the behaviorally assessed pain perception (negative values indicate pain inhibition; positive values indicate pain facilitation).

| | Pain modulation in VAS ratings of pain intensity | | | | | | Pain modulation in behavioral measure (°C) | | | | | |
| --- | --- | --- | --- | --- | --- | --- | --- | --- | --- | --- | --- | --- |
| | placebo | | levodopa | | naltrexone | | placebo | | levodopa | | naltrexone | |
| | n=28 | | n=27 | | n=28 | | n=28 | | n=27 | | n=28 | |
| Outcome | *M* | *SD* | *M* | *SD* | *M* | *SD* | *M* | *SD* | *M* | *SD* | *M* | *SD* |
| win | –7.31 | 21.51 | –12.98 | 23.54 | –10.09 | 23.79 | –0.09 | 0.64 | –0.14 | 0.66 | –0.05 | 0.74 |
| lose | 12.21 | 21.12 | 13.29 | 20.48 | 12.26 | 22.27 | 0.03 | 0.59 | 0.03 | 0.54 | 0.06 | 0.68 |

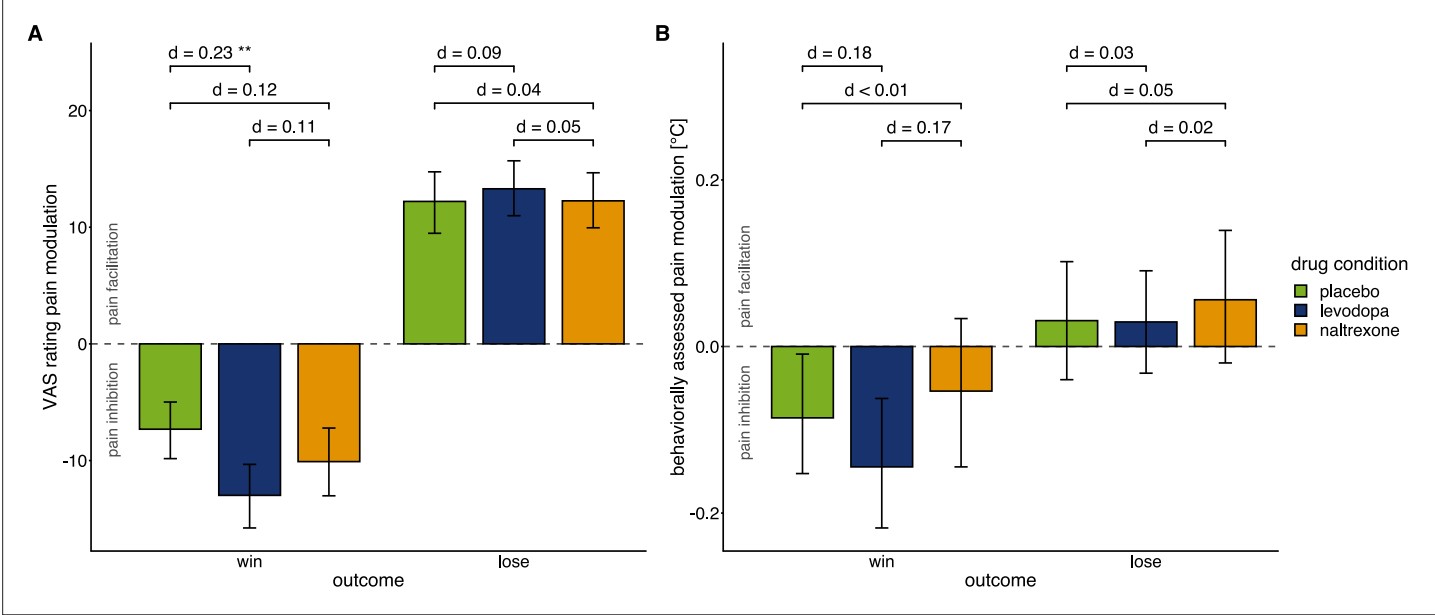

**Figure 3.** Effects of drug manipulation on endogenous pain modulation. Effects of drug manipulation on endogenous pain modulation assessed by VAS ratings of pain intensity (**A**) and behaviorally assessed pain perception (**B**) after winning and losing in the wheel of fortune game, respectively (placebo: n=28, levodopa: n=27, naltrexone: n=28). Bars show group level means and error bars show 95% confidence interval of the group level mean. d indicates the standardized effect-size after controlling for random effects and residual variance. While the temporal order of sessions did affect pain modulation (**Figure 3—figure supplement 1**), measures of pain sensitivity, that were not experimentally manipulated (**Figure 3—figure supplement 2**), and measures of mood (**Figure 3—figure supplement 3**) did not significantly differ between drug conditions. For individual effects of the drug manipulations on endogenous pain modulation see **Figure 3—figure supplement 4**.

The online version of this article includes the following figure supplement(s) for figure 3:

**Figure supplement 1.** Effects of temporal order of sessions on endogenous modulation.

**Figure supplement 2.** Baseline pain sensitivity and neutral condition of the wheel of fortune task.

**Figure supplement 3.** Mood ratings.

**Figure supplement 4.** Individual level effects of drug manipulations on endogenous modulation.

effect of active relief on perception was significantly larger in the levodopa condition compared to the placebo condition (post-hoc comparison p=0.007, *d*=0.23; **Figure 3A**). No significant difference was found for the naltrexone compared to the placebo condition (p=0.252, *d*=0.12). Endogenous modulation did not significantly differ between the levodopa and the naltrexone condition (p=0.368, *d*=0.11). Endogenous pain facilitation induced by actively receiving pain increases assessed with VAS ratings did not significantly differ between any drug conditions (all post-hoc comparisons p*'s*>0.591).

## Behaviorally assessed pain perception

In contrast to the placebo condition, participants showed significantly less behaviorally assessed sensitization in active compared to passive trials when obtaining pain relief under levodopa (pairwise comparison test vs. control: p=0.020, *d*=0.11; **Figure 2E**) consistent with an extension of pain-inhibitory effects of winning pain relief through to the underlying tonic pain stimulus. Under naltrexone, test and control trials did not significantly differ in the behaviorally assessed pain perception (**Figure 2F**) as for the placebo condition. Across drugs, behaviorally assessed pain modulation did not significantly differ between placebo, levodopa, and naltrexone (interaction 'drug × outcome': $F_{(2, 1592.73)}$=1.87, p=0.154; **Figure 3B**).

## Levodopa and naltrexone influence relief reinforcement learning in the wheel of fortune task

To investigate whether pain relief gained in active relief seeking was associated with an impact on choice related to reinforcement learning, one of the 2 choices in the wheel of fortune was associated

with a fixed 75% chance of winning pain relief ($choice_{high\ prob}$ while the other choice only had a 25% chance to win pain relief ($choice_{low\ prob}$). Participants were not informed of these probabilities in advance. We tested if the proportion of choices of the more rewarding option was higher in the last two out of five blocks of four test trials each of the game, when the subjects already had the chance to explore and learn the different outcome probabilities.

Participants selected the color of the wheel of fortune associated with a higher likelihood for winning relief in 64% (SD = 28%) of trials in the placebo condition, consistent with a reinforcement learning effect. Thus, participants chose the color associated with the higher likelihood for winning above chance ($\chi^2(1)=6.64$, p=0.010) on a group level, indicating successful learning.

However, participants' performance significantly differed between the placebo and the drug conditions (main effect of 'drug': $\chi^2(2) = 11.89$, p=0.003). In contrast to the placebo condition (post-hoc comparison p<0.001), under levodopa and under naltrexone participants' choices did not significantly differ from chance (post-hoc comparisons p's>0.759). Correspondingly, post-hoc comparisons show that choice behavior significantly differed in the placebo compared to the levodopa condition (p=0.015) and compared to the naltrexone condition (post-hoc comparison p=0.004), while choices did not significantly differ between levodopa and naltrexone (post-hoc comparison p=0.915; *Figure 4*). This shows that both, dopamine and opioids, may have an influence on relief-related learning and choice.

In an additional exit interview at the end of each session, participants were asked whether they believed that one color of the wheel was associated with a higher chance of winning pain relief. The proportion of participants who reported this color correctly was not above chance (binomial test: p's>0.5; placebo: 50%, levodopa: 37%, naltrexone: 39.3%). Nevertheless, participants' belief whether one color of the wheel of fortune task was associated with a higher chance of winning or not significantly influenced their choices (p<0.001) and this influence on choices, and thus on learning, depended on the drug condition (interaction 'drug × belief': $F(2) = 6.91$, p=0.032). Group effects of successful learning, i.e. selecting the color with a higher chance of winning, were driven by participants who were able to report this association ($p\ (choice_{high\ prob}|correct\ belief) = 0.737$, $p\ (choice_{high\ prob}|false\ or\ no\ belief) = 0.545$; post-hoc comparison: p=0.007) under placebo and naltrexone (p's<0.001) but not under levodopa (p=0.922). This suggests that successful decision-making was associated with contingency awareness in our task. However, the current data does not allow a conclusion on whether this contingency awareness was a prerequisite for or a consequence of successful learning here.

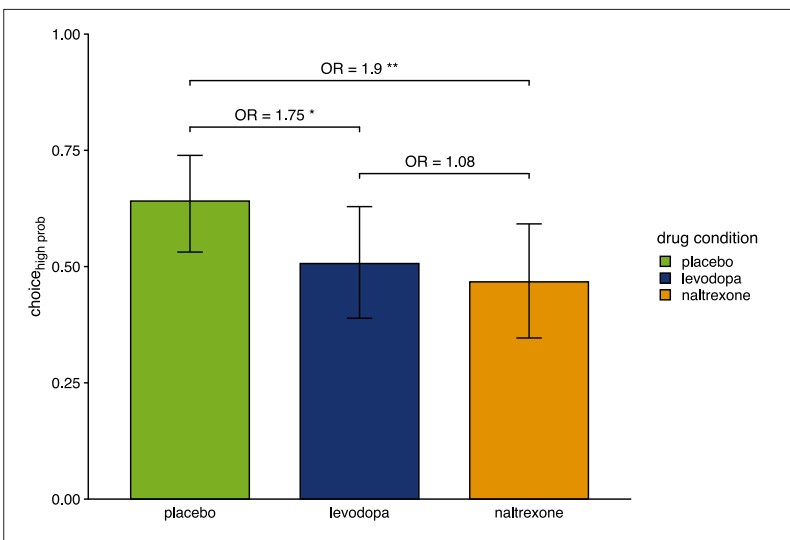

**Figure 4.** Proportion of choices of the color associated with a higher chance of winning pain relief. Bars show group level means and error bars show 95% confidence interval of the group level mean (placebo: n=28, levodopa: n=27, naltrexone: n=28). OR indicates odds ratios as effect size of estimated effects between drugs. *p<0.05, ** p<0.01.

The online version of this article includes the following figure supplement(s) for figure 4:

**Figure supplement 1.** Individual level effects of drug manipulations on choice behavior.

## Unpredictability and endogenous pain modulation

We next tested whether outcome unpredictability, indicated by reward prediction errors of winning (pain relief) and losing (pain increase) in the game, was associated with endogenous pain modulation, and whether this association differed between drugs. Using hierarchical Bayesian modeling, we fit reinforcement learning models that captured the update of expected values for choice through outcomes of the wheel of fortune (*Glimcher, 2011*), with a drift diffusion process as the choice rule to participants' choice and reaction time data. The best predictive accuracy was found for model 4 that used an individually scaled outcome sensitivity, and a sigmoid function to map expected values for the two choices to the drift rate of the diffusion process (*Table 2*; see Materials nd methods, section *Estimation of prediction errors and their role in endogenous pain modulation* for details on parametrization of reward learning models).

Posterior predictive simulations from the best-fitting model appropriately describe the observed choices (*Figure 5*). However, none of the model parameters could exclusively explain the differences between levodopa and naltrexone compared to placebo: the 95% highest density intervals (HDI) for the difference between all group level parameters of the drug effect enclosed zero (see *Figure 5—figure supplement 1*).

Prediction errors estimated by using subject level parameters of the model showed a significant main effect for the prediction of endogenous pain modulation indicated by VAS ratings ($F(1, 1600.3) = 452.9$, $p < 0.001$). A negative estimate of the prediction error ($\beta_{PE} = -0.36$) indicates that outcomes that are better than expected (positive prediction errors, which occur when receiving relief) were related to increased relief perception (pain inhibition). Conversely outcomes that are worse than expected (negative prediction errors, occurring with pain increases) were associated with increased pain facilitation (*Figure 6*). In other words, the more unexpected the relief, the greater the perception of that relief; and the more unexpected the pain increase, the greater the perception of that pain.

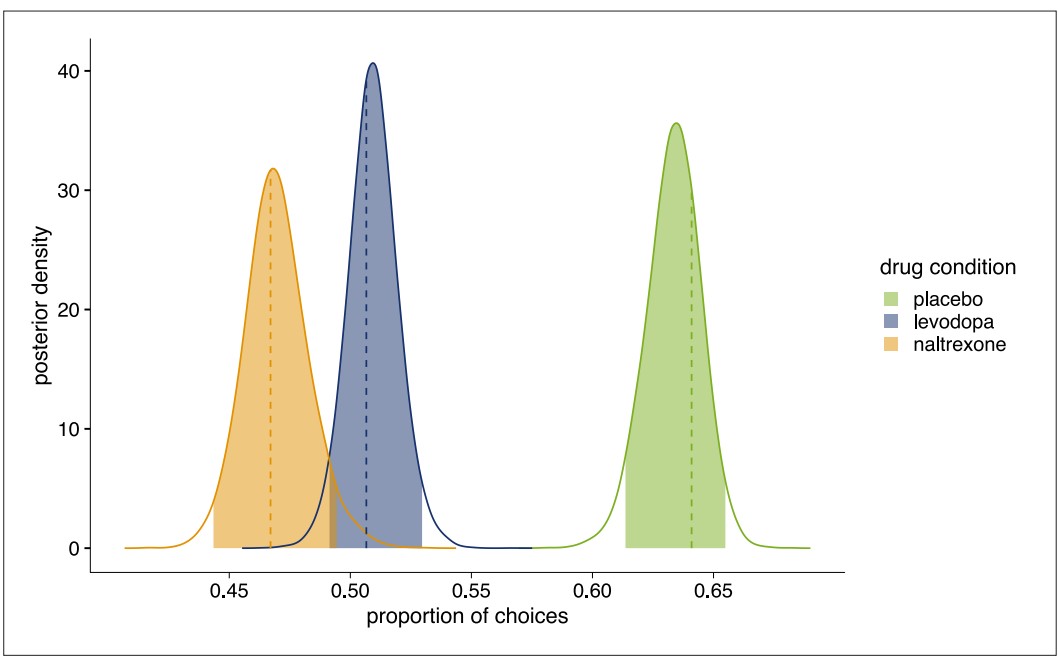

**Figure 5.** Posterior distributions of the proportion of choices in favor of $choice_{high\ prob}$. Colored areas show 95% highest density interval ($HDI_{95}$). Dashed lines indicate observed proportion of choices in favor of $choice_{high\ prob}$. Placebo (n=28): $p\left(choice_{high\ prob}\right) = 0.641$, $HDI_{95} = [0.614, 0.655]$, posterior (p-value ($pp$)=0.320); levodopa (n=27): $p\left(choice_{high\ prob}\right) = 0.507$, $HDI_{95} = [0.491, 0.530]$, p=0.679; naltrexone (n=28): $p\left(choice_{high\ prob}\right) = 0.467$, $HDI_{95} = [0.443, 0.494]$, p=0.611. *Figure 5—figure supplement 1* shows comparison of drug conditions for each parameter of winning model 4.

The online version of this article includes the following figure supplement(s) for figure 5:

**Figure supplement 1.** Differences of the posterior distributions of group level parameters for the main effect of drug in model 4.

**Table 2.** Model comparison.
Models are ordered by their expected log pointwise predictive density (*ELPD*). *ELPD*_diff: difference to the *ELPD* of winning model 4. *se(ELPD_diff)*: standard error of the difference in *ELPD*.

| Model | ELPD | ELPD_diff | se(ELPD_diff) |
|---|---|---|---|
| Model 4 | –837.71 | 0 | 0 |
| Model 3 | –845.44 | –7.73 | 1.51 |
| Model 2 | –997.33 | –159.62 | 15.77 |
| Model 1 | –998.33 | –160.62 | 15.95 |

The effect of prediction errors on pain modulation showed a significant interaction with the drug condition ($F$(2, 1599.5)=7.529, p<0.001). Post-hoc analysis confirmed that the negative linear relationship significantly differed from zero for all conditions (p's<0.001), but this relationship was significantly stronger for levodopa compared to placebo (p<0.001) with no significant differences for naltrexone compared to placebo (p=0.083). Overall, this shows that relief is enhanced to unpredictability, and this effect is sensitive to dopamine.

Estimated prediction errors also showed a significant main effect for the prediction of behaviorally assessed pain modulation ($F$(1, 1602.1)=9.00, p=0.003), with a negative estimate ($\beta_{PE} = -0.06$) suggesting that sensitization decreased with smaller prediction errors. No significant interaction with of prediction error with drug conditions was found for behaviorally assessed pain perception (interaction 'PE × drug': $F$(2, 1600.1)=0.96, p=0.384).

## Novelty seeking is linearly associated with increased endogenous pain modulation by pain relief under levodopa

Previous data suggest that endogenous pain inhibition induced by actively winning pain relief is associated with a novelty seeking personality trait: greater individual novelty seeking is associated with greater relief perception (pain inhibition) induced by winning pain relief (*Becker et al., 2015*). Similar to these results, we found here that endogenous pain modulation, assessed using self-reported pain intensity, induced by winning was associated with participants' scores on novelty seeking in the NISS

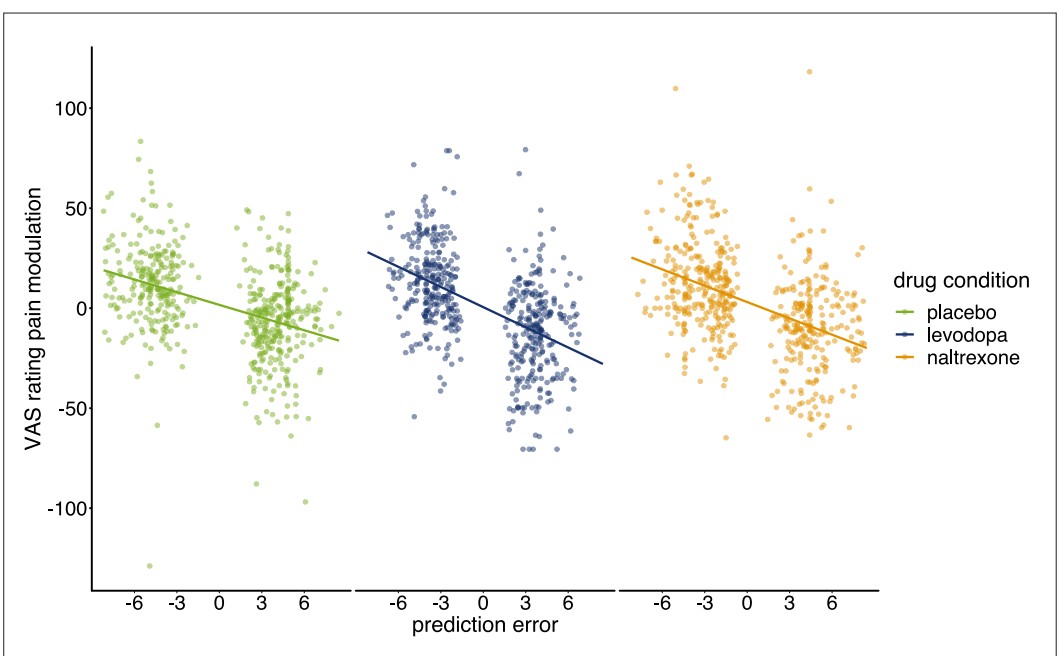

**Figure 6.** Pain modulation in VAS ratings predicted by prediction errors for each condition. Regression lines indicate prediction from the mixed effects model with predictors 'PE', 'drug', and their interaction (placebo: n=28, levodopa: n=27, naltrexone: n=28).

questionnaire (Need Inventory of Sensation Seeking; *Roth and Hammelstein, 2012*; subscale 'need for stimulation' (NS)), although this correlation failed to reach statistical significance after correction for multiple comparisons using Bonferroni-Holm method ($r=-0.412$, $p=0.073$). A similar association between novelty seeking and endogenous pain modulation was found in the levodopa condition ($r=-0.551$, $p=0.013$). More importantly, the higher a participants' novelty seeking score in the NISS questionnaire, the greater the levodopa-related endogenous pain modulation when winning compared to placebo (NISS NS: $r=-0.483$, $p=0.034$, *Figure 7*). In contrast, higher novelty seeking scores were not correlated with stronger pain modulation induced by winning in the naltrexone condition ($r=0.153$, $p=0.381$) and the naltrexone induced change in pain modulation showed no significant association with novelty seeking ($r=0.239$, $p=0.499$). Pain modulation after losing was not associated with novelty seeking in placebo ($r=0.083$, $p=0.866$), levodopa ($r=-0.164$, $p=0.783$), or naltrexone ($r=0.405$, $p=0.133$).

No significant correlations with NISS novelty seeking score were found for behaviorally assessed pain modulation in the placebo, levodopa and naltrexone conditions during pain relief or pain increase ($|r|$'s<0.35, $p$'s>0.238). Similarly, the difference in pain modulation during pain relief or pain increase between the levodopa and the placebo condition and between the naltrexone and the placebo condition did also not correlate with novelty seeking ($|r|$'s<0.22, $p$'s>0.576).

## Discussion

The results show that (i) the perception of relief is sensitive to endogenous modulation during motivated behavior, (ii) this modulation scales with the informational content of the relief, being enhanced when relief is actively controllable, more unexpected, and especially in high trait novelty seeking individuals, (iii) this information-specific modulation is sensitive to manipulation of dopamine signaling, with no significant effects of the manipulation of opioidergic signaling on endogenous modulation; (iv) however, both dopaminergic and opioidergic signaling have an influence on relief-seeking, which may be at least in part dissociable from relief perception. Overall, this shows that dopaminergic signaling is involved in a fundamental component of the endogenous modulation of pain relief.

Theories of the endogenous modulation of pain propose that one of the reasons that pain is modulated is to optimize motivational behavior, in terms of responding, learning, and decision-making

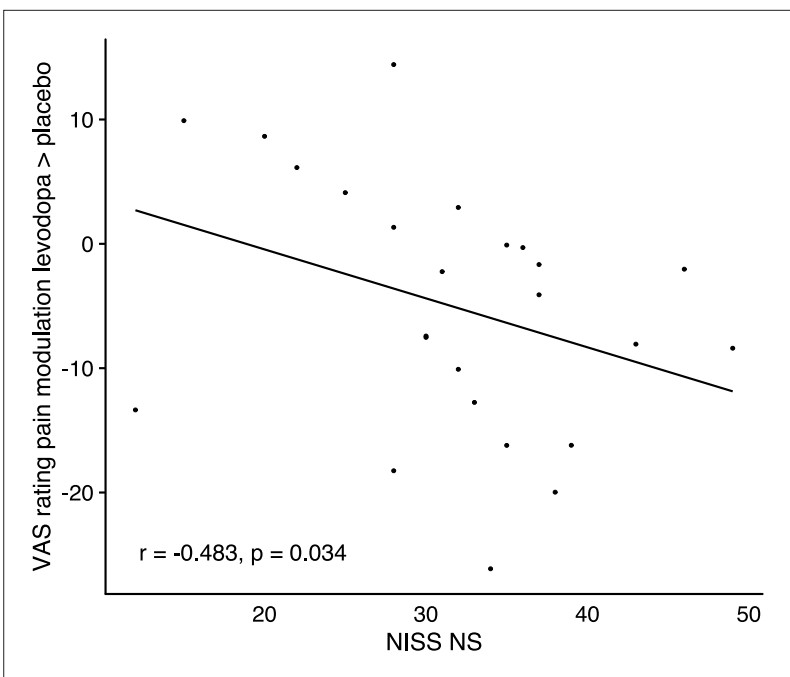

**Figure 7.** Correlation of changes in endogenous pain modulation induced by winning pain relief under levodopa compared to placebo with individuals' scores on the 'need for stimulation' subscale of the NISS questionnaire, n=24.

(*Fields, 2018*; *Seymour, 2019*). That is, pain is increased in situations in which it has a more important role in shaping behavior – for instance when it directs a change in behavior (instrumentally controllable), when it is partly unpredictable (i.e. contains new information), and in otherwise dangerous contexts. This theory centralizes the functional role of pain as a signal for behavioral control that is concerned with the *prospective* control of behavior. In principle, this can be extended as a potential account for the modulation of relief, because the offset of pain is also important as a control signal for guiding behavior, one which occurs in the context of an ongoing noxious event, such as an injury of some sort (*Zhang et al., 2018*). We have previously found preliminary evidence of this, by showing that relief perception is enhanced by active controllability (*Becker et al., 2015*). Here, we intended to test this more precisely, by looking at the role of controllability as well as unpredictability, and also compare to the modulation of phasic increases in tonic pain.

We also set an additional prediction, in that we expected to find that modulation by information content would be greater in novelty-seeking individuals (*Becker et al., 2015*). This is because novelty seeking describes an explicit information-seeking tendency, in which new information is explored with the potential to lead to knowledge of better outcomes that can be exploited in the future (*Wittmann et al., 2008*). This illustrates the common basis for intrinsic motivation for novelty and information-seeking for exploitable benefit, and hence we can predict that high trait novelty seekers might be more sensitive to information that occurs through relief outcomes.

Overall, all three predictions were largely borne out by the data: relief perception as measured by VAS ratings was enhanced by controllability, unpredictability and showed a medium sized – although not significant – association with the individual novelty-seeking tendency, consistent with the hypothesis that relief is sensitive to the exploitable information it carries. This provides the first clear formal framework for understanding a key component of relief perception. The principles for controllability and unpredictability also extended to increases in pain, consistent with the notion that increases in tonic pain act in a similar way to phasic pain operating from a pain-free baseline.

## Effects of pharmacological manipulations on endogenous perceptual modulation

Both dopamine and opioids are implicated in relief processing, although their precise roles remain unclear. We found endogenous relief modulation here was modulated by enhanced dopamine availability induced by the intake of levodopa. Importantly, all three core aspects of informational-sensitivity were modulated by levodopa: active controllability, unpredictability, and association with novelty seeking. These findings also illustrate potential parallels with the previous observation of endogenous pain inhibition by extrinsic monetary reward co-occurring with experimental pain (*Becker et al., 2013*). In this context, monetary reward represents an independent and potentially competing incentive, and when this co-occurs with pain, it means that optimal responding may require suppression of pain responses, especially innate responses that could interfere with reward acquisition. In both cases, the common principle may be the active 'decision' by the pain system to tune incoming pain signals to optimize behavior. Levodopa significantly strengthened the association of endogenous modulation of the perception with the extent to which nociceptive input was unpredictable or surprising, suggesting that dopamine is involved in a fine-grained perceptual modulation that may help to control immediate reactions to pain related cues as well as to optimize prospective behavior. Moreover, our results support the view that effects of dopamine on pain perception are not unidirectional but depend on the motivational and the informational value of nociceptive signals. Note that the personality trait of novelty seeking has also been associated with enhanced dopaminergic activity due to lower midbrain (auto)receptor availability (*Leyton et al., 2002*; *Savage et al., 2014*; *Zald et al., 2008*), which further supports a general role for dopamine in information-sensitive behavior (*Kakade and Dayan, 2002*; *Vellani et al., 2020*).

The role of dopamine in pain relief in the context of reinforcement is supported by findings of increased dopamine release induced by pain relief in the Nucleus accumbens of rats (*Navratilova et al., 2012*; *Xie et al., 2014*). Dopamine release was related to the development of conditioned place preference that could be blocked by dopamine antagonists (*Navratilova et al., 2012*). Further, *Navratilova et al., 2015b* showed that dopamine release in the Nucleus accumbens and conditioned place preference in response to pain relief depend on opioidergic signaling: both were blocked by opioid antagonism in the anterior cingulate cortex, an area encoding pain aversiveness.

In contrast to our hypothesis, pharmacologically blocking opioid receptors using naltrexone did not reduce modulation endogenous pain inhibition in this task. The doses and methods used here are comparable to those used in other contexts which have identified opioidergic effects. Using positron emission tomography *Weerts et al., 2013* found a blockage of μ-opioid receptors of more than 90% by 50 mg of naltrexone (p.o.) in humans given repeatedly over 4 days. In addition, effects on behavioral functions have been reported with comparable doses that support the efficacy of the opioidergic manipulation. *Chelnokova et al., 2014* found attenuating effects of 50 mg naltrexone (p.o.) on wanting as well as liking of social rewards, implicating the involvement of endogenous opioids in the processing of rewarding stimuli. The same dose was also found to attenuate reward directed effort exerted in a value-based decision-making task (*Eikemo et al., 2017*). Moreover, 50 mg of naltrexone (p.o.) have been shown to reduce endogenous pain inhibition induced by conditioned pain modulation (*King et al., 2013*). Thus, based on the literature we assume that the opioidergic manipulation was effective in this study, although we do not have a direct manipulation check of this pharmacological manipulation. Despite its effectiveness in blocking endogenous opioid receptors, the effect of naltrexone on reward responses was found to be small (*Rabiner et al., 2011*). Hence, a lack of power may have limited our chances to find such effects in the present study.

In humans, *Sirucek et al., 2021* showed that perception of passively received pain relief was reduced by blocking of opioid receptors. However, in that task, received pain relief did not carry behaviorally relevant information, in contrast to the present task. Further, *Sirucek et al., 2021* asked participants to rate the perceived pain relief and its pleasantness, while in the current study participants were asked to rate perceived intensity of the stimulation at certain time points in each trial. Perceived pain relief was therefore estimated indirectly by differences in these ratings. Increased opioid activity in the anterior cingulate cortex has been shown to be associated with selectively decreased pain aversiveness with unaltered sensory pain components (*Gomtsian et al., 2019*; *Maruyama et al., 2018*; *Navratilova et al., 2015b*). In contrast, the present study aimed at quantifying the effect of controllability on the relief perception, with these methods possibly not capturing the effects of opioid blockade on positive affective quality components of the relief experience. Future studies need to address the question whether the affective dimension of enhanced relief perception by rewarding pain relief may be reduced by opioid blockade with the sensory component possibly being unaffected.

Although we did not assess the affective component of the relief experience, we implemented two outcome measures that are assumed to capture independent aspects of the pain experience: VAS ratings indicate perception of phasic changes (outcomes), while the behavioral measure indicates perceptual within-trial sensitization or habituation in response to the tonic stimulation within each trial. We found enhanced endogenous modulation by controllability and unpredictability in VAS ratings, in line with the view that endogenous modulation enhances behaviorally relevant information. In contrast, the within-trial sensitization did not differ between the active and passive conditions under placebo. In a previous study using a similar experimental paradigm *Becker et al., 2015* found a reduction of within-trial sensitization after pain relief outcomes by active controllability. Compared to this study we implemented smaller changes in stimulation intensity as outcomes in the wheel of fortune (–3 °C vs –7 °C for pain relief), potentially explaining the differential results.

## Effects of pharmacological manipulations on relief-seeking behavior

One key difference in the current version of the wheel of fortune task, compared to the previous version described in *Becker et al., 2015*, is that participants' choices had a non-random association with outcomes that is this was a true instrumental (operant) contingency between actions and outcomes. This allowed us to assess a basic measure of learning – whether subjects are able to learn to select more frequently the option with the better (75% chance of relief) over worse (25% chance of relief) outcome. That both levodopa and naltrexone conditions were associated with a reduction of the frequency of choosing the better option, indicates that signals mediated by both neurotransmitters may be involved in choice. However, the data argue against a simple transposition of experienced relief (measured by VAS) into decision value, which for a stationary task such as this, should lead to more deterministic actions in the levodopa condition but no effect under naltrexone compared to placebo. The association of explicit contingency awareness and choice in our task illustrates the fact that multiple decision systems ('model-based' and 'model-free') might be involved in even simple instrumental tasks, and hence that more sophisticated task manipulations

are needed to decompose these different components (*Langdon et al., 2018*). However, our key finding is that there is at least a simple dissociation between the drug effects on experienced relief and decision-making.

Such dissociation may be due to differential involvement of dopamine and endogenous opioids in different yet interacting aspects of reward and punishment processing. Dopamine has been related to instrumental learning due to its prominent role in mediating reward and aversive prediction errors (*Glimcher, 2011*; *Matsumoto and Hikosaka, 2009*; *Schultz, 2007*; *Schultz, 2016*). Correspondingly, effects of dopaminergic modulation on value-based decision making and brain activity related to reward prediction errors in the Nucleus accumbens have been reported (*Pessiglione et al., 2006*). On the other hand, impaired learning functions under dopaminergic medication are known from research in Parkinson's disease (*Breitenstein et al., 2006*; *Pizzagalli et al., 2008*; *Santesso et al., 2009*; *Vo et al., 2016*) and have been attributed to dopamine overstimulation (*Cools et al., 2001*; *Vaillancourt et al., 2013*). Others argued that dopamine overstimulation does not impair learning of associations or reward expectations, but only the transfer to overt actions (choice behavior; *Beeler, 2012*; *Beeler et al., 2010*). Accordingly, *Kroemer et al., 2019* found reduced model-free control of choice behavior under levodopa (i.e. a decrease in direct reinforcement of actions by rewards) while both, neural reward prediction error signals and also model-based learning remained unaffected. Given the involvement of multiple decision systems in our task increased dopamine availability might have led to increased explorative behavior. When exploration is favored over exploitation choice behavior is less driven by values learned from the prior reward history. At the same time, dopamine has also been implicated in motivational aspects (incentive salience) of reward processing (*Berridge et al., 2009*; *Smith et al., 2011*; *Tindell et al., 2005*). Hence, dopamine may have increased motivational drive and related facilitation of pain modulation in the present task, while at the same time increased dopamine availability may have reduced the expression of prior reward learning (*Beeler, 2012*). Opioids have been related to both, incentive salience and the hedonic value of rewards (*Berridge et al., 2009*; *Meier et al., 2021*). In humans, bidirectional manipulations have shown that opioid agonism increases while opioid antagonism decreases "wanting" (i.e. incentive salience) as well as "liking" of attractive faces (*Chelnokova et al., 2014*). The same mechanism was also shown for the effort to work for and the response bias for higher monetary rewards indicating that opioid manipulations affect motivation but also choice behavior (*Eikemo et al., 2017*). These findings suggest that inhibition of opioidergic activity by blocking endogenous opioid receptors could impair reward processing (independent of effects of endogenous opioids on pain modulation), and hence, explain why participants did not develop a preference for the choice option that was associated with a higher chance to win pain relief under naltrexone in the present task.

## Implication and perspectives

Because the mechanisms underlying learning from pain and pain relief and their recursive influence on pain perception may contribute to the development and maintenance of chronic pain, it is crucial to better understand the roles of dopamine and endogenous opioids in these mechanisms. Accordingly, bidirectional manipulations of both transmitter systems should be used in future studies to better characterize their respective roles in shaping behavior and perception.

The data may have clinical implications. Reward learning has recently been shown to play a role in the transition of acute to chronic pain with a specific pattern of Nucleus accumbens activity in response to a cue predicting pain relief being predictive for chronification (*Löffler et al., 2022*). This makes pain relief processing a potential leverage point for prevention strategies. Although levodopa or dopamine agonists are not generally used as analgesics in the clinical management of chronic pain, it may be that they could have a potential adjuvant role in management programs, for example when used in the context of rehabilitation strategies that aim to harness endogenous control mechanisms. It is also worth noting that Parkinson's disease has a well-recognized association with chronic pain, beyond that which can be explained by motor effects, and in keeping with a potential core role for dopamine in the pathogenesis of chronic pain in some contexts (*Beiske et al., 2009*).

In summary, our study shows that dopamine has a core role in pain relief information processing, by which it modulates the way in which information tunes the modulation of pain to meet motivational demands.

## Materials and methods

### Participants

Thirty healthy volunteers (16 female, 14 male; age: mean = 27.1 years; SD = 7.9 years) participated in this study. Exclusion criteria were present pain or pain conditions in the last 12 months, mental disorders, excessive gambling, substance abuse behaviors, alcohol consumption of 100 ml or more of alcohol per week, regular night shifts, or sleep disorders. The power estimation was based on the design and the finding of a medium effect size in a previous study using a comparable version of the wheel of fortune game without pharmacological interventions (*Becker et al., 2015*). The a priori sample size calculation for an 80% chance to detect such an effect at a significance level of yielded a sample size of 28 participants (estimation performed using GPower [*Faul et al., 2007*; version 3.1] for a repeated-measures ANOVA with a three-level within-subject factor). The study was approved by the Ethics Committee of the Medical Faculty Mannheim, Heidelberg University, and written informed consent was obtained from all participants prior to participation according to the revised Declaration of Helsinki (*World Medical Association, 2013*).

### Testing sessions

Each participant performed three testing sessions on separate days. Each session comprised a pharmacological intervention and a wheel of fortune game to assess modulation of reward-induced endogenous pain modulation by the interventions. Participants received in one session levodopa to transiently increase the availability of dopamine, in one session the opioid receptor antagonist naltrexone to block opioid receptors, and in one session a placebo for control. To ensure complete washout of the drugs, the testing sessions were separated by at least 2 days (plasma half-life for levodopa: 1.4 hrs *Nyholm et al., 2012*; plasma half-life for naltrexone: 8 hrs [*Wall et al., 1981*]). After obtaining written consent in the first testing session, participants were familiarized with the thermal stimuli, the rating scale, and the wheel of fortune game to decrease unspecific effects of novelty and saliency. In each testing session, the thresholding and scaling procedures for individual adjustments of the stimulation intensities started approximately 60 min after drug intake and were performed immediately prior to playing the wheel of fortune game.

### Thermal stimulation

All heat stimuli were applied using a 25x50 mm contact thermode (SENSELab—MSA Thermotest, SOMEDIC Sales AB, Sweden). The baseline temperature was set to 30 °C. Rise and fall rates of the temperature were set to 5 °C/s. All thermal stimuli were applied to the inner forearm of participants' non-dominant hand after sensitization of the skin using 0.075% topical capsaicin cream to allow for potent pain relief as reward and pain increase as punishment without the risk of skin damage (*Becker et al., 2015*; *Gandhi et al., 2013*). By activating temperature-dependent TRPV1 (vanilloid transient receptor potential 1) ion channels capsaicin as the active ingredient of chili pepper induces heat sensitization (*Holzer, 1991*). To ensure that the entire area of thermal stimulation during the wheel of fortune game was sensitized the cream was applied to an area on the forearm exceeding the area of stimulation by about 1 cm on each side. After 20 min, the capsaicin cream was removed (*Dirks et al., 2003*; *Gandhi et al., 2013*) and the thermode was applied. If participants reported the baseline temperature of the thermode (30 °C) as painful because of the preceding sensitization this temperature was lowered until it was perceived as non-painful, which was needed in 8 out of 83 sessions (3 placebo sessions, 1 levodopa session, and 4 naltrexone sessions) that were finally entered into the analysis (see below). The temperature was decreased to 28 °C (1 placebo session, 4 naltrexone sessions) or 26 °C (1 placebo session, 1 levodopa session). The need to lower the baseline temperature was not significantly different between drug conditions (Fisher's exact test, *P*=0.52).

### Determination of stimulation intensities

Participants' heat pain threshold and heat pain tolerance were assessed using the method of limits three times prior to the wheel of fortune game. The temperature of the thermode increased from baseline with 1 °C/s. Participants were instructed to press the left button of a three-button computer mouse when the pain threshold was reached. The respective temperature was recorded while the temperature further increased. Participants were instructed to press the button again when the pain tolerance threshold was reached. The respective temperature was recorded and the temperature

immediately returned to baseline. The arithmetic mean of the temperatures corresponding to the recorded pain threshold and tolerance in the three trials was used as an estimate of the individual heat pain threshold and heat pain tolerance, respectively.

After this threshold and tolerance assessment, an adjustment procedure resembling a staircase method was implemented to determine the stimulation intensities in the wheel of fortune game. Participants received heat stimuli of 20 s duration and continuously rated the perceived intensity of these stimuli on a computerized visual analogue scale (VAS) ranging from 'no sensation' (0) over 'just painful' (100) to 'most intense pain tolerable' (200) (*Becker et al., 2013*; *Villemure et al., 2003*) while the stimuli where presented. The VAS scale was presented on the screen with a red marker that could be moved along the scale. Participants adjusted the position of the marker by pressing the right or left mouse button. The marker moved into the respective direction until participants released the button. The participants were instructed to adjust their rating whenever a change in their perception occurred. Participants could adjust their ratings as long as the VAS scale was presented on the screen. The temperature of the first trial was set to the mean of the previously determined pain threshold and tolerance. If the rating at the end of the stimulation was outside a range of 150±10 on the VAS, the temperature for the next trial was adjusted according to the difference to a target rating of 150. This adjustment was determined by multiplying the difference (150 – current rating) by 0.02 and adding the result in °C to the previous temperature. Further, temperature increases between trials were limited to a maximum of 0.5 °C to avoid overshooting of ratings. The procedure was repeated until a rating between 140 and 160 on the VAS was achieved, aiming at a temperature perceived as moderately painful. The corresponding temperature was used as the stimulation intensity in the wheel of fortune game.

## Wheel of fortune game

A wheel of fortune game, adapted from a previously established version (*Becker et al., 2015*), was used to provide participants with the possibility of winning pain relief. The game comprised three types of trials: *test* trials, in which participants played the wheel of fortune game and received pain relief or pain increases according to the outcome of the game; *control* trials, in which participants did not play the game, but received pain relief or pain increases as in the test trials; and *neutral* trials, in which participant did not play the game and no pain relief or pain increases were implemented. A trial always started with an increase of the temperature to the previously determined tonic pain stimulation intensity. When the stimulation intensity was reached, participants were instructed to memorize the temperature perceived at this moment (*Figure 1*). After this memorization interval, participants were presented with a wheel of fortune display that was divided into three sections of equal size but different color.

In the *test* trials, participants were asked to select one of two colors (pink or blue) of the wheel by pressing a corresponding button (left or right) on the mouse. This started the wheel spinning (4.3 s) until it stopped on either the blue or pink section. When the wheel came to a stop and the pointer of the wheel indicated the color the participant had chosen, the stimulation temperature decreased with the aim to induce pain relief (win condition). If the pointer indicated the color the participant had not chosen, the temperature was increased (lose condition). In the *control* trials, participants had to press a black button unrelated to the sections of the wheel of fortune using the middle mouse button, after which the wheel started spinning as in the test trials. In contrast to the test trials, no pointer was displayed in the control trials and the wheel stopped at a random position. After the wheel came to a stop, the stimulation temperature decreased or increased, resembling the course of stimulation in the test trials, but without winning or losing. By this procedure, nociceptive input in test and control trials was kept the same, allowing to test specifically for endogenous pain modulation induced by winning and losing in the wheel of fortune game.

In *neutral* trials, participants had to press a black button, as in the control trials, after which the wheel also started spinning. In these neutral trials, the pointer of the wheel always landed the third color of the wheel (white), which could not be selected in test trials, and the stimulation temperature did not change. Neutral trials were used to estimate changes in pain perception occurring over the course of the experiment due to habituation or sensitization independent of the outcomes of the wheel of fortune game.

The participants were instructed that there were two types of trials: trials in which they could choose a color to bet on the outcome of the wheel of fortune and trials in which they had no choice. Specifically, they were told that in the first type of trials they could use the left and right mouse button, respectively, to choose between the pink and blue section of the wheel of fortune. Participants were further instructed that if the wheel lands on the color they had chosen they will win, that is that the stimulation temperature will decrease, while if the wheel lands on the other color, they will lose, that is that the stimulation temperature will increase. For the second type of trials, participants were instructed that they could not choose a color, but were to press a black button, and that after the wheel stopped spinning the temperature would by chance either increase, decrease, or remain constant.

After the interval of the temperature change (in the test trials: outcome of the wheel), participants rated the perceived intensity of the current temperature using the same VAS as described above (*Figure 1*). After this rating, participants had to adjust the stimulation temperature themselves to match the temperature they had memorized at the beginning of the trial. This self-adjustment operationalizes a behavioral assessment of perceptual sensitization and habituation within one trial (*Becker et al., 2011*; *Becker et al., 2015*; *Kleinböhl et al., 1999*). Participants adjusted the temperature using the left and right button of the mouse to increase and decrease the stimulation temperature. The behavioral measure was calculated as the difference in temperatures in the memorization interval at the beginning of each trial minus this self-adjusted temperature at the end of each the trial. Positive values, that is self-adjusted temperatures lower than the stimulation intensity at the beginning of each trial, indicate perceptual sensitization, while negative values indicate habituation. After this behavioral assessment, the stimulation temperature went back to baseline and after a short break (5 s) the next trial started.

In total, the wheel of fortune game comprised of 45 trials, split into five blocks. Each block consisted of 4 *test* and 4 *control* trials followed by one *neutral* trial, resulting in 20 test trials in which participants chose a color, 20 control trials, and 5 neutral trials. *Test* and *control* trials were presented in a predefined, pseudorandomized sequence. In contrast to the previous version of the wheel of fortune (*Becker et al., 2015*), the outcome of the wheel occurred with certain likelihood to allow for learning to optimize the outcomes of the wheel of fortune. One of the colors (pink or blue) was associated with a 75% chance of winning, while the other was associated with a 25% chance of winning (counterbalanced across participants and testing sessions). If participants did not select a color in the test trials, the neutral outcome (white) of the wheel was displayed and the temperature did not change. The temperature changes in the *control* trials (pain relief or increase) were matched to the outcomes of the *test* trials to ensure that the same number of pain relief and pain increase trials were presented in test and control trials.

Pain relief was implemented by a reduction of the stimulation intensity of 3 °C and pain increase was implemented by a rise of 1 °C. The magnitude of these temperature steps was determined and optimized in pilot experiments with the aim of inducing potent pain relief and pain increase without inducing ceiling and floor effects.

Although the main focus of the study was to test different effects on pain relief as implemented in win trials and their corresponding control trials with a decrease in nociceptive input, lose trials and their complementing control trials were crucial to the experimental design. First, for playing the game lose trials were an integral part because of the implemented likelihood for winning which necessarily needs to be accompanied by the chance of losing. Additionally, the risk of losing was thought to increase the participants' engagement in the game, which in turn was expected to enhance the motivated state induced by playing the wheel of fortune game. Second, they allowed for testing whether pain modulation was driven by controllability or unspecific effects such as arousal and distraction in test compared to control trials of the wheel of fortune game (*Becker et al., 2015*).

All experimental procedures involving thermal stimulation were controlled by custom-programmed Presentation scripts (Presentation software, Version 17.0, http://www.neurobs.com) providing instructions and other visual cues on a computer screen in front of the participants (code available at https://osf.io/5xjt9).

## Pharmacological manipulations

Participants ingested in one testing session *levodopa*, in another *naltrexone*, and in another a placebo (microcrystalline cellulose), following a double-blind, cross-over design with counterbalanced order. Levodopa is an amino acid precursor of dopamine leading to a transient systemic increase of dopamine availability. To inhibit peripheral synthesis of dopamine from levodopa, the single dose of 150 mg levodopa (p.o.) was combined with 62.5 mg of a benserazide to prevent peripheral side effects such as nausea (*Rinne et al., 1975*). Naltrexone is an opioid receptor antagonist with predominant receptor binding affinity at μ-opioid receptors together with a lower binding affinity at κ-opioid receptors and a much lower affinity at δ-opioid receptors (*Raynor et al., 1994*). Participants received a single dose of 50 mg naltrexone (p.o.) that has been shown to induce more than 90% receptor blockade (*Weerts et al., 2013*).

After drug intake, a waiting period of one hour started. This waiting time was chosen based on peak plasma concentrations of levodopa and naltrexone at approximately 1 hr to 1.5 hr after ingestion (*Nyholm et al., 2012*; *Wall et al., 1981*). At the end of each testing session, participants indicated whether they thought that they had received the placebo or one of the drugs (response alternatives: 'placebo', 'levodopa', 'naltrexone', or 'don't know') to test for potential unblinding.

## Questionnaire and exit interview

Novelty seeking as personality trait was assessed using the Need Inventory of Sensation Seeking (NISS; *Roth and Hammelstein, 2012*). The NISS consists of the sub-scales *Need for Stimulation* (NS) and *Avoidance of Rest* (AR). We used the NS subscale as a measure for novelty seeking as it reflects the 'need for novelty and intensity' (*Roth and Hammelstein, 2012*). Before playing the wheel of fortune game the affective state of subjects was assessed using computerized versions of the Self-Assessment Manikin (SAM; *Bradley and Lang, 1994*; *Lang, 1980*) and a German version (*Krohne et al., 1996*) of the Positive And Negative Affect Scale (PANAS; *Watson et al., 1988*). At the of each session, an exit interview was performed, asking for the following information: (1) which drug participants believed to have ingested; (2) if participants believed that choosing one of the two colors was associated with a higher chance to win pain relief; (3) whether participants perceived a difference between test and control trials; (4) whether participants had the impression that the stimulation temperature at the beginning of each trial varied across trials; (5) whether participants had problems indicating their perception on the VAS scale; and (6) whether participants had problems readjusting the initial temperature. Participants gave first yes/no answers and then were asked to specify their answers using open-ended questions.

## Statistical analysis

For the statistical analysis, two participants were excluded, one participant due to the failure to comply with experimental procedures and one due to technical failure of the equipment. For one additional participant, data of one session (levodopa) are missing due to drop-out. Thirty-two out of 3735 single trials of all the remaining sessions were not recorded due to technical failures. In 42 trials, participants did not press a button within the respective interval in the wheel of fortune game. These trials were excluded from the analyses. Note that the NISS questionnaire was missing for two additional subjects due to initial issues at the beginning of the data collection.

To test if blinding was successful we fit a mixed effects logistic regression with the subjects' assumption on the ingested drug (as reported in the exit interview, see above) being correct as dependent variable. We used 'drug' as a fixed factor and to account for repeated measures we modeled a random intercept for each subject. Post-hoc general linear hypothesis tests were used to compare estimated proportions of correct assumptions against chance.

To confirm that the manipulation of the motivated state (test vs. control trials) of the participants by playing the wheel of fortune game did induce pain modulation as intended in each session, we analyzed the VAS ratings and the behavioral pain measure as outcome measures separately for each session with 'trial type' and 'outcome' as well as their interaction as fixed effects. Here, the interaction effect indicates that active controllability in test trials has different effects for pain relief and pain increase. We used post-hoc tests to confirm that pain intensity was lower in the test compared to the control condition in win trials and equal or higher in the test compared to control condition in lose trials. We used Bonferroni-Holm method for family-wise error correction of these post-hoc tests across

drug conditions. To account for the repeated measures design, we modeled a random intercept for each participant and a random slope for outcome of the wheel within each participant.

To obtain an estimate for endogenous pain modulation in each test trial, we subtracted the mean value of all control trials of either the pain relief or the pain increase trials from the value of the winning or losing test trials separately for each session for both the VAS ratings and the behavioral pain measure. Using these differences, negative values indicate pain inhibition and positive values indicate pain facilitation. Estimates for pain modulation were analyzed using linear mixed model procedures with the fixed factors 'drug' (levodopa, naltrexone, placebo), 'outcome' (win, lose), 'order' of sessions (1, 2, 3), and their interaction separately for ratings and behaviorally assessed pain perception as dependent variables. The factor 'session number' was added to control for effects of temporal order independent of the drug manipulation that was found to influence pain modulation (see *Figure 3— figure supplement 1*). Other factors such as baseline pain perception or mood did not affect pain modulation and were not included in further analysis. To account for the repeated measures design we modeled a random intercept for each participant and a random slope for outcome within each participant.

Unbeknown to the subjects, one of the colors in the wheel of fortune was associated with a higher chance to win pain relief. To test whether participants learned to select this color from the implemented reward contingencies we looked at choice behavior in the last 2 blocks of trials only. In this latter phase of the task subjects already had the chance to explore differences in outcomes associated with their choices and were thought show exploitation if they had learned about the contingency. We fitted a mixed effects logistic regression with the subjects' choices as dependent variable. For a single session, we fit an intercept only model where the intercept represents the group level estimate for the probability to choose the color associated with a higher chance of winning pain relief ($choice_{high\ prob}$). Drug was used as an additional within-subject factor when testing for differences among levodopa, naltrexone, and placebo. To account for repeated measures, we modeled a random intercept for each subject. To assess the effect of the subjects' belief about which color was associated with a higher chance to win pain relief (as reported in the exit interview, see above) we added the factor 'belief' (either 'correct belief' or 'false or no belief') to this model.

To test whether endogenous pain modulation due to winning pain relief was related to participants' personality trait of novelty seeking, pain modulation represented by the differences between test and control trials in the wheel of fortune in VAS ratings and the behaviorally assessed pain perception of the placebo and the levodopa condition were correlated with the NISS NS scores. To test further whether increases in pain modulation induced by levodopa were associated with novelty seeking, differences in pain modulation between the levodopa and placebo session were also correlated with the NISS NS scores. Before calculating these correlations, multivariate outliers were tested using a chi-square test on the squared Mahalanobis distance using an of 0.025 (*Filzmoser, 2016*), leading to the exclusion of one value for the correlation with the difference of pain modulation between the levodopa and placebo session.

The significance level was set to 5% for all analyses. All statistical analyses were performed using statistical computing software R version 3.5.3 (*R Development Core Team, 2019*). Mixed model analyses were performed using the *lme4* package (*Bates et al., 2015*). All linear mixed models were estimated using restricted maximum likelihood. Kenward-Roger correction as implemented in the *lmerTest* package (*Kuznetsova et al., 2017*) was used to calculate test statistics and degrees of freedom to account for the sample size. For general linear mixed effects models Wald $\chi^2$ was calculated using *car* package (*Fox and Weisberg, 2011*). Post-hoc tests and effect sizes were calculated on estimated marginal means using the *emmeans* package (*Lenth, 2020*) where appropriate. Effects sizes were calculated by dividing the difference in marginal means by the pooled standard deviation of the random effects and the residuals providing an estimate for the underlying population (*Hedges, 2007*). Tukey adjustment was used to account for multiple comparisons in post-hoc tests.

## Estimation of prediction errors and their role in endogenous pain modulation

To analyze how mechanisms of instrumental learning contribute to the observed choice behavior and how this related to reward-induced pain modulation, we fitted reinforcement learning (RL) models to participants' choices in test trials of the wheel of fortune game. Such models were initially formulated

for associative learning (**Bush and Mosteller, 1951**; **Rescorla and Wagner, 1972**) and adapted for instrumental learning (**Sutton and Barto, 1998**). RL models assume that actions are chosen based on the expected outcome. Learning is described as the adaptation of expectations based on experiences. Thus, learning is driven by the discrepancy between a present expectation and the obtained outcome, namely the *prediction error*. The speed of adaption of the expectation is described by the *learning rate*, which defines the exponential decay of the influence of previous outcomes on the currently present expectation. For trial-by-trial instrumental learning paradigms, the update of the expectation of an outcome related to a given action (in the present study: choice in the wheel of fortune) is operationalized by calculating the expected value $Q$ of a choice as follows:

$$Q_{choice,\,t+1} = Q_{choice,\,t} + \eta \times \delta_t \tag{1}$$

where $Q_{choice}$ is the reward expectation for a given choice, $t$ denotes the trial, $\eta$ is the learning rate, and $\delta_t$ is the prediction error in trial $t$. The learning rate $\eta$ determines the speed of adaption; the higher $\eta$ the more is the expectation influenced by recent compared to former experiences. Since updating of expectations has been shown to differ dependent on the sign of the prediction error (**Fontanesi et al., 2019**; **Gershman, 2015**; **Pedersen et al., 2017**), we modeled independent learning rates for positive ($\eta_+$) and negative ($\eta_-$) prediction errors:

$$Q_{choice,\,t+1} = Q_{choice,\,t} + \eta_+ \times \delta_t, \ \ if \ \delta_t > 0 \tag{2}$$
$$Q_{choice,\,t+1} = Q_{choice,\,t} + \eta_- \times \delta_t, \ \ if \ \delta_t \leq 0 \tag{3}$$

The prediction error as the difference between the actual and the expected outcome in trial $t$ is formulated as follows:

$$\delta_t = R_t - Q_{choice,\,t} \tag{4}$$

with $R_t$ as the outcome of the choice in trial $t$.

In the wheel of fortune game, outcomes were implemented as changes in stimulation intensities. Accordingly, $R_t$ was positive (+1) for temperature decreases in win trials or negative (–1) for temperature increases in lose trials. The formula shown above assumes a constant outcome sensitivity. To capture potential modulation of the outcome sensitivity, we implemented a scaled outcome sensitivity so that the reward in trial $t$ was multiplied by an individual scaling factor $\rho$ yielding a scaled prediction error:

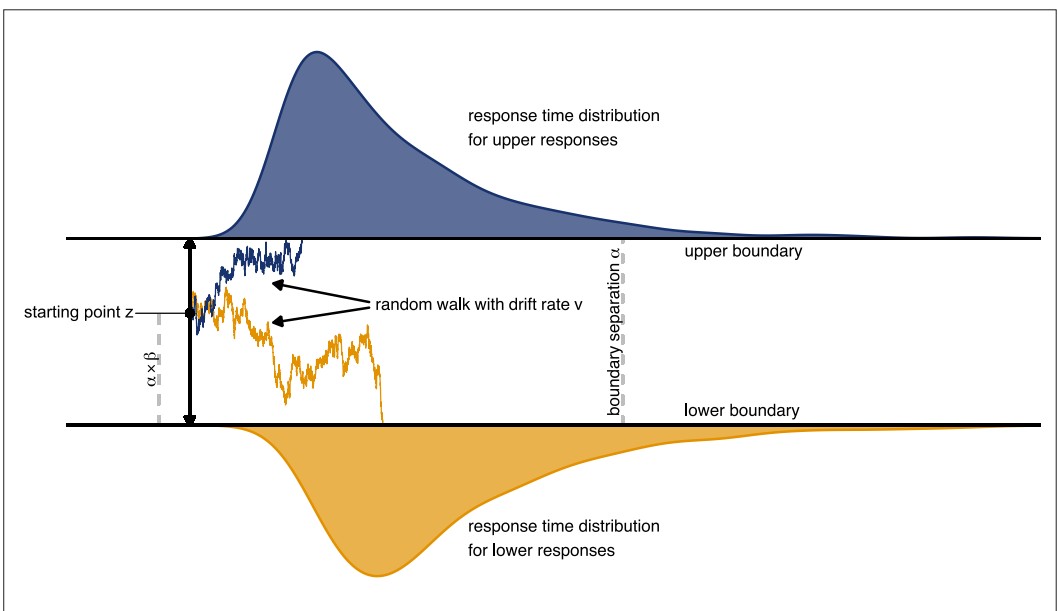

**Figure 8.** Schematic depiction of the drift diffusion process. Accumulation of evidence starts at point $z$ which is defined by the a-priori bias $\beta$ and the boundary separation $\alpha$. Noisy evidence is integrated over time (represented by sample paths in blue and orange, for upper and lower boundary choices, respectively).

$$\delta_t = (\rho \times R_t) - Q_{choice, t} \tag{5}$$

Q values were initiated to zero and calculated separately for choices of the color associated with a higher chance to win pain relief ($Q_{high\,prob}$) and choices of the color associated with a lower chance to win pain relief ($Q_{low\,prob}$).

While RL models traditionally used a softmax choice rule (*Daw and Doya, 2006*; *Luce, 1959*), recent studies on value-based decision making have implemented variants of the drift diffusion model (*Ratcliff, 1978*; *Ratcliff and Rouder, 1998*) to map expected values to choices (*Fontanesi et al., 2019*; *Pedersen et al., 2017*; *Peters and D'Esposito, 2020*). The drift diffusion model describes decisions as accumulation of noisy evidence for two choice options until a predefined threshold, representing either of the two options, is reached. Such drift diffusion models take response times (*RT*) of decisions into account and model mathematically cognitive processes underlying the decision process. *Figure 8* depicts such a decision process. The range between the decision boundaries is represented by the boundary separation parameter $\alpha$. Higher values of $\alpha$ lead to slower but more accurate decisions, that is, $\alpha$ represents the speed vs. accuracy tradeoff. The position of the starting point $z$ between the boundaries is determined by a priori biases $\beta$ toward one of the two options. This parameter $\beta$ represents the relative distance of $z$ between the boundaries. It can range from 0 to 1 where a value of 0.5 indicates no bias, values below indicate a bias for the lower choice and values above 0.5 a bias for the upper choice. The non-decision time $\tau$ describes time needed for processes that are unrelated to the decision process (e.g. stimulus processing). Correspondingly, the reaction time is defined as $RT = \tau + decision\ time$. Acquisition of evidence starts from the starting point $z$ at time $\tau$ as a random walk. The slope of this random walk is determined by the drift rate $\nu$ and a decision is made when either the upper or lower boundary is reached. Higher drift rates result in faster and more accurate decisions. The probability of the *RT* when choosing option $x$ can then be calculated using the Wiener first-passage time distribution (*Ratcliff, 1978*):

$$RT(x) = Wiener(\alpha, \tau, \beta, \nu) \tag{6}$$

where *Wiener*() returns the probability that is chosen with the observed *RT*.

Most variants of reward learning models that use the drift diffusion process as a choice rule replace the constant drift rate by an individually scaled difference of expected values for the both options (*Fontanesi et al., 2019*; *Pedersen et al., 2017*; *Peters and D'Esposito, 2020*). Thus, the drift rate $\nu_t$, varies across trials as a function of the difference between expected values of the two choice options that in the wheel of fortune corresponded to $Q_{high\,prob}$ and $Q_{low\,prob}$, respectively. We implemented a linear mapping of the difference in expected values like *Pedersen et al., 2017* where this difference is multiplied by the scaling factor $\nu$:

$$\nu_t = (Q_{high\,prob} - Q_{low\,prob}) \times \nu \tag{7}$$

As an alternative scaling method, we implemented a non-linear function as suggested by *Fontanesi et al., 2019* in which the scaled difference in expected values is mapped to the drift rate using a sigmoid function, which more closely resembles the non-linear mapping of the softmax function:

$$\nu_t = S((Q_{high\,prob, t} - Q_{low\,prob, t}) \times \nu) \tag{8}$$

where $S(x)$ is defined as:

$$S(x) = \frac{2 \times \nu_{max}}{1 + e^{-x}} - \nu_{max} \tag{9}$$

With that, $\pm \nu_{max}$ defines the upper and lower limit of the drift rate, respectively, while the shape or slope of the sigmoid function depends on the scaled difference of expected values.

In summary, we combined different parameterizations of the outcome sensitivity (static or scaled) and the mapping of expected values to the drift rate (linear or sigmoidal) into different models (*Table 3*).

We used hierarchical Bayesian modeling to fit the reward learning models to the choices of the participants in the test trials. Hierarchical models estimate group and individual parameters simultaneously to mutually inform and constrain each other, which yields reliable estimates for both, individual and group level parameters (*Gelman et al., 2013*; *Kruschke, 2014*). Posterior distributions of the

**Table 3.** Model specification.

Models 1–4 were defined using different combinations of parameters for reward sensitivity and the mapping of expected values to the drift rate. A 'static' reward sensitivity means that pain increase and pain decrease were defined as –1 and 1, respectively (see *Equation 4*). A 'scaled' outcome sensitivity means that pain decrease was defined as $-\rho$ and pain decrease as $\rho$ (see *Equation 5*). A 'linear' drift rate mapping means that the drift rate $\nu_t$ for each trial was defined as the difference of expected values multiplied by $\nu$ (see *Equation 7*). A sigmoid mapping of the drift rate means that $\nu_t$ was defined by a sigmoid function bounded at $\pm\nu_{\max}$ . (see *Equation 8* und *Equation 9*). All models included two learning rates ($\eta_+$ , $\eta_-$), the non-decision time $\tau$, the boundary separation $\alpha$, and the a priori bias $\beta$.

| Model | Outcome sensitivity | Drift rate mapping |
|---|---|---|
| Model 1 | static | linear |
| Model 2 | scaled | linear |
| Model 3 | static | sigmoid |
| Model 4 | scaled | sigmoid |

parameters were estimated using Hamiltonian Monte Carlo sampling with a No-U-Turn sampler as implemented in the probabilistic language Stan (*Carpenter et al., 2017*) via its *R* interface *rstan* (*Stan Development Team, 2020*). For each model parameter, we included a global intercept and the main effect of drug (levodopa, naltrexone, placebo). Both, intercept and main effect were allowed to vary for each participant and we modeled a correlation of individual terms for the drug effect across participants to account for repeated measures. We used a non-centered parameterization to reduce dependency between group and individual level parameters (*Betancourt and Girolami, 2015*). Therefore, both intercept and drug effect were defined by their location (group level effect), scale, and error (individual effects) distributions. A logistic transformation was applied to the learning rate ($\eta_+$ , $\eta_-$) and a priori bias ($\beta$) parameters to restrict values to the range of $\begin{bmatrix} 0, \ 1 \end{bmatrix}$ . The location parameters for the intercept of the learning rate were given standard normal priors ($\mathcal{N}(0, \ 1)$) and the scale of these parameters were given half-normal priors ($\mathcal{HN}(0, 1)$). The location of the drug effect on learning rate parameters were also given standard normal priors while the scale was given a half-normal prior of $\mathcal{HN}(0, 0.1)$ to prevent allocation of high prior density at the edges of the range after logistic transformation, resulting in an almost flat prior. The location parameter for the intercept of the a-priori bias was given a normal prior of $\mathcal{N}(0, 0.5)$ and the scale was given a half-normal prior $\mathcal{HN}(0, 0.1)$ . The location parameter for the drug effect was given a normal prior of $\mathcal{N}(0, 0.5)$ and the scale was given a half-normal prior of $\mathcal{HN}(0, 0.1)$ . To ensure that the non-decision time ($\tau$) was bounded to be lower than the reaction time the parameter was equivalently transformed to the range [0,1] and multiplied with each subject's individual minimum reaction time in a given session. Priors were the same as for the learning rate, that is yielding a flat prior after transformation. We used an exponential transformation to constrain the reward sensitivity parameter ($\rho$), the boundary separation ($\alpha$, drift rate scaling factor ($\nu$), and the boundary of the drift rate ($\nu_{\max}$) to be greater than 0. The location of the global intercept was given a normal prior of $\mathcal{N}(0.1, 0.1)$ for the reward sensitivity, a normal prior of $\mathcal{N}(0, 0.1)$ for the boundary separation, a normal prior of $\mathcal{N}(0.2, 0.2)$ for the drift rate, and a normal prior of $\mathcal{N}(0.5, 0.2)$ for the drift rate boundary. For the exponentially transformed parameters, the scale of the global intercept was given a half-normal prior of $\mathcal{HN}(0, 0.1)$, the location of the drug effect was given a normal prior of $\mathcal{N}(0, 0.5)$, and the scale of the drug effect was given a half-normal prior of $\mathcal{N}(0, 0.1)$. Individual effects for the intercept as well as for the drug effect were all given standard normal priors. The correlation matrix of individual drug-level effects for each parameter was given a LKJ prior (*Lewandowski et al., 2009*) of $\mathcal{LKJcorr}(1)$. All models were run on four chains with 4000 samples each. The first 1000 iterations were discarded as warm-up samples for each chain. The convergence of chains was confirmed by the potential scale reduction factor $R$ .

The fitted models were compared for their best predictive accuracy using *K*-fold cross-validation (*Vehtari et al., 2017*). For the cross-validation, we split data into $k = 10$ subsets with each subset containing data of 2–3 participants and calculated the expected log pointwise predictive density (*ELPD*) based on simulations for each hold-out set $y_k$ using parameters estimated from re-fitting

the model to the training data set $y_{(-k)}$. We calculated $ELPDs$, their differences, and the standard error of the differences using the $R$ package $loo$ (**Vehtari et al., 2020**). A higher $ELPD$ indicates a better predictive accuracy. Such a better predictive accuracy was assumed if the difference in $ELPD$ ($ELPD_{diff}$) for two models was at least two times the standard error of that difference ($se\,(ELPD_{diff})$).

For the best fitting model, we performed posterior predictive checks by simulating replicated data sets from posterior draws. As the test statistic for the posterior predictive check, we examined the proportion of choices in favor of the option associated with a higher chance to win pain relief ($choice_{high\,prob}$) in the last 2 blocks of the wheel of fortune game and compared the proportions observed in this data to the distribution of proportions found in the simulated data sets.

From the best fitting model, we used group level estimates for the main effect of 'drug' to compare model parameters between drug conditions using the 95% highest density interval (HDI) of the difference of their posterior distributions.

The means of individual parameter posterior distributions were used to estimate prediction errors for single trials. To test whether these prediction errors predict endogenous pain modulation induced by the wheel of fortune task, we used linear mixed models with the fixed factors 'prediction error' and 'drug', and their interaction. A random intercept for each subject was included to account for repeated measures. Separate models for VAS ratings and behaviorally assessed pain perception as dependent variables were calculated.

## Additional information

### Funding

| Funder | Grant reference number | Author |
| --- | --- | --- |
| Baden-Württemberg Stiftung | Postdoctoral Fellowship for Leading Early Career Researchers | Susanne Becker |
| Universität Heidelberg | Olympia Morata Program | Susanne Becker |
| Swiss National Science Foundation | PRIMA grant | Susanne Becker |
| Deutsche Forschungsgemeinschaft | Collaborative Research Centres SFB1158 B03 and B07 | Herta Flor |
| Wellcome Trust | Senior Research Fellowship (214251/Z/18/Z) | Ben Seymour |
| Versus Arthritis | Research Award 21537 | Ben Seymour |
| Ministry of Science and ICT, South Korea | Technology Planning & Evaluation IITP grant 2019-0-01371 | Ben Seymour |
| Deutsche Forschungsgemeinschaft | Reinhart Koselleck Project Fl 156/41-1 | Herta Flor |
| Swiss National Science Foundation | PR00P1_179697/1 | Susanne Becker |

The funders had no role in study design, data collection and interpretation, or the decision to submit the work for publication. For the purpose of Open Access, the authors have applied a CC BY public copyright license to any Author Accepted Manuscript version arising from this submission.

### Author contributions

Simon Desch, Conceptualization, Data curation, Formal analysis, Investigation, Visualization, Methodology, Writing – original draft, Writing – review and editing; Petra Schweinhardt, Conceptualization, Supervision, Writing – review and editing; Ben Seymour, Conceptualization, Methodology, Writing – review and editing; Herta Flor, Conceptualization, Resources, Supervision, Writing – review and editing; Susanne Becker, Conceptualization, Resources, Formal analysis, Supervision, Funding

acquisition, Methodology, Writing – original draft, Project administration, Writing – review and editing

## Author ORCIDs
Simon Desch (ID) http://orcid.org/0000-0001-5863-643X
Ben Seymour (ID) http://orcid.org/0000-0003-1724-5832
Susanne Becker (ID) http://orcid.org/0000-0002-5681-4084

## Ethics

Human subjects: The study was approved by the Ethics Committee of the Medical Faculty Mannheim, Heidelberg University, (approval reference 2014-504N-MA) and written informed consent was obtained from all participants prior to participation according to the revised Declaration of Helsinki (World Medical Association, 2013).

## Decision letter and Author response
Decision letter https://doi.org/10.7554/eLife.81436.sa1
Author response https://doi.org/10.7554/eLife.81436.sa2

---

# Additional files

## Supplementary files
• MDAR checklist

## Data availability

Behavioral and questionnaire data is available as csv file at the project's Open Science Framework page (osf.io/5xjt9).

The following dataset was generated:

| Author(s) | Year | Dataset title | Dataset URL | Database and Identifier |
|---|---|---|---|---|
| Desch S, Schweinhardt P, Seymour B, Flor H, Becker S | 2022 | Endogenous modulation of pain relief | https://osf.io/5xjt9 | Open Science Framework, 5xjt9 |

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
