## [Editor Report]

This is an important paper that is of interest to researchers interested in the psychological and neurochemical mechanisms of pain and pain relief. It shows that the perception of pain relief is modulated by controllability, surprise and novelty seeking. Moreover, these modulations are influenced by dopaminergic but not by opioidergic manipulations. These findings are supported by convincing evidence.

---

## [Decision Letter]

**Decision letter after peer review:**

Thank you for submitting your article "Endogenous modulation of pain relief: evidence for dopaminergic but not opioidergic involvement" for consideration by *eLife*. Your article has been reviewed by 3 peer reviewers, including Markus Ploner as the Reviewing Editor and Reviewer #1, and the evaluation has been overseen by Joshua Gold as the Senior Editor. The following individuals involved in review of your submission have agreed to reveal their identity: Choong-Wan Woo (Reviewer #2); Siri Leknes (Reviewer #3).

Essential revisions:

1) There is no direct evidence that the opioidergic manipulation has been effective. Efficacy should either be directly demonstrated, inferred from previous studies or at least critically discussed.

2) The study was powered to detect medium effect sizes, based on a previous study using a version of the same gambling task. The within-subjects design is very appropriate here and enhances the statistical power, however the authors should clarify in the manuscript that they did not power the study to detect a number of the effects probed, e.g. the effects of the pharmacological manipulations, which could plausibly be small based on at least some of the previous literature. This means null effects in particular should be interpreted with caution. This would also mean that the authors might shift the focus away from stating that dopaminergic but not opioidergic manipulations influence pain relief to stating that dopaminergic manipulations have stronger effects than opioidergic manipulations. This shift should then be substantiated by a direct statistical contrast of dopaminergic and opioidergic manipulations. Alternatively, the authors could provide direct evidence that opoidergic manipulations have no effects by using Bayesian statistics. This would allow the authors to stick to the statement that dopaminergic but not opioidergic manipulations influence pain relief.

3) The authors run a large number of tests. They test a number of different – and very interesting – hypotheses so this is not unreasonable, but a bit more information on error control would be useful. Note that I do not recommend an overall Bonferroni or related approach, but perhaps family-wise correction for each hypothesis test would be appropriate.

4) The effects were found in one (pain intensity ratings) but not the other (behaviorally assessed pain perception) outcome measure. This weakens the findings and should at least be critically discussed.

5) The instructions given to the participants should be specified. Moreover, it is essential to demonstrate that the instructions do not yield differences in other factors than controllability (e.g., arousal, distraction) between test and control trials. Otherwise, the main interpretation of a controllability effect is substantially weakened.

6) The blinding assessment does not rule out that the volunteers perceived the difference between placebo on the one hand and levodopa/naltrexone on the other hand. It is essential to directly show that the participants were not aware of this difference.

7) The effects of novelty seeking have been assessed in the placebo and the levodopa but not in the naltrexone conditions. This should be explained. Assessing novelty seeking effects also in the naltrexone condition might represent a helpful control condition supporting the specificity of the effects in the naltrexone condition.

8) The writing of the manuscript is sometimes difficult to follow and should be simplified for a general readership. Sections on the information-processing account of endogenous modulation in the introduction (lines 78-93), unpredictability and endogenous pain modulation in the results (lines 278-331) are quite extensive and add comparatively little to the main findings. These sections might be shortened and simplified substantially. Moreover, providing a clearer structure for the discussion by adding subheadings might be helpful.

9) Effect sizes are generally small. This should be acknowledged and critically discussed. Moreover, effect sizes are given in the figures but not in the text. They should be included to the text or at least explicitly referred to in the text.

10) Given that the Results section comes before the Methods section, it would be helpful to include some method and experimental design-related information crucial for the understanding of the results in the Results section. For example, how long was the thermal stimulus? What was the baseline temperature? etc. Maybe this information can be included in the caption of Figure 1.

11) t would be helpful if the authors could provide their prior hypotheses on the drug effects. It could be a little bit confusing that the goal of using these drugs given that levodopa is a precursor of dopamine, whereas naltrexone is the opioid antagonist, i.e., the effects on the target neurotransmitters seem the opposite. Then, I wondered if the authors expected to see the opposite effects, e.g., levodopa enhances pain relief, while naltrexone inhibits pain relief, or to see similar effects, e.g., both enhance pain relief. Clarifying which direction of expected effects would be helpful for novice readers.

12) On the "Behaviorally assessed pain perception" results in Figures 2D-F, I wonder why the results for the "pain increase" were still positive. Were the y values on the plots the temperature that participants adjusted (i.e., against the temperature right before the temperature adjustment)? or are the values showing the differences from the baseline (i.e., against the baseline temperature)?

13) Some more methods information should be included in the Results section for ease of reading, e.g. which statistical model is used to produce each result, whether session order was included etc.

*Reviewer #1 (Recommendations for the authors):*

– Line 598. How were continuous pain ratings performed?

*Reviewer #2 (Recommendations for the authors):*

No further comments.

*Reviewer #3 (Recommendations for the authors):*

The study was powered to detect medium effect sizes, based on a previous study using a version of the same gambling task. The within-subjects design is very appropriate here and enhances the statistical power, however the authors should clarify in the manuscript that they did not power the study to detect a number of the effects probed, e.g. the effects of the pharmacological manipulations, which could plausibly be small based on at least some of the previous literature. This means null effects in particular should be interpreted with caution. Relatedly, the authors run a large number of tests. They test a number of different – and very interesting – hypotheses so this is not unreasonable, but a bit more information on error control would be useful. Note that I do not recommend an overall Bonferroni or related approach, but perhaps family-wise correction for each hypothesis test would be appropriate.

Some more methods information should be included in the Results section for ease of reading, e.g. which statistical model is used to produce each result, whether session order was included etc.

---

## [Author Response]

Essential revisions:1) There is no direct evidence that the opioidergic manipulation has been effective. Efficacy should either be directly demonstrated, inferred from previous studies or at least critically discussed.

We agree that we cannot provide direct evidence on the effectiveness of the opioidergic manipulation in our study. However, previous literature strongly suggests that a dose of 50 mg naltrexone (p.o.) is effective in blocking μ-opioid receptors in humans. Using positron emission tomography, Weerts et al. (2013) found a blockage of μ-opioid receptors of more than 90% with 50 mg naltrexone (p.o.) although given repeatedly 4 days in a row. In addition, convincing effects on behavioral functions have been reported with comparable doses that support the efficacy of the opioidergic manipulation. For example, Chelnokova et al. (2014) found attenuating effects of 50 mg naltrexone (p.o.) on wanting as well as liking of social rewards, implicating the involvement of endogenous opioids in the processing of rewarding stimuli. The same dose was also found to attenuate reward directed effort exerted in a value-based decision-making task (Eikemo et al., 2017). Moreover, 50mg of naltrexone (p.o.) have been shown to reduce endogenous pain inhibition induced by conditioned pain modulation (King et al., 2013) and to reduce the perceived pleasantness of pain relief (Sirucek et al., 2021). Thus, based on the available literature we assume the effectiveness of our opioidergic manipulation. A corresponding reasoning including a note of caution on the of the lack of a direct manipulation check of the opioidergic manipulation can be found in the manuscript in the Discussion (page 21 lines 505ff):

“The doses and methods used here are comparable to those used in other contexts which have identified opioidergic effects. Using positron emission tomography, Weerts et al. (2013) found a blockage of opioid receptors of more than 90% by 50 mg of naltrexone (p.o.) in humans given repeatedly over 4 days. In addition, effects on behavioral functions have been reported with comparable doses that support the efficacy of the opioidergic manipulation. Chelnokova et al. (2014) found attenuating effects of 50 mg naltrexone (p.o.) on wanting as well as liking of social rewards, implicating the involvement of endogenous opioids in the processing of rewarding stimuli. The same dose was also found to attenuate reward directed effort exerted in a value-based decision-making task (Eikemo et al., 2017). Moreover, 50 mg of naltrexone (p.o.) have been shown to reduce endogenous pain inhibition induced by conditioned pain modulation (King et al., 2013). Thus, based on the literature we assume that the opioidergic manipulation was effective in this study, although we do not have a direct manipulation check of this pharmacological manipulation. Despite its effectiveness in blocking endogenous opioid receptors, the effect of naltrexone on reward responses was found to be small (Rabiner et al., 2011). Hence, a lack of power may have limited our chances to find such effects in the present study.”

2) The study was powered to detect medium effect sizes, based on a previous study using a version of the same gambling task. The within-subjects design is very appropriate here and enhances the statistical power, however the authors should clarify in the manuscript that they did not power the study to detect a number of the effects probed, e.g. the effects of the pharmacological manipulations, which could plausibly be small based on at least some of the previous literature. This means null effects in particular should be interpreted with caution. This would also mean that the authors might shift the focus away from stating that dopaminergic but not opioidergic manipulations influence pain relief to stating that dopaminergic manipulations have stronger effects than opioidergic manipulations. This shift should then be substantiated by a direct statistical contrast of dopaminergic and opioidergic manipulations. Alternatively, the authors could provide direct evidence that opoidergic manipulations have no effects by using Bayesian statistics. This would allow the authors to stick to the statement that dopaminergic but not opioidergic manipulations influence pain relief.

We agree with the reviewers that the power may not have been sufficient to detect potentially small effects of the pharmacological manipulations. The power calculation was based on the design and the medium effect size found in a previous study using a comparable experimental procedure for assessing pain-reward interactions (Becker et al., 2015). To acknowledge this weakness, we clarified in the manuscript the description of the a priori sample size calculation as follows (page 26, lines 645ff):

“The power estimation was based on the design and the finding of a medium effect size in a previous study using a comparable version of the wheel of fortune game without pharmacological interventions (Becker et al., 2015). The a priori sample size calculation for an 80% chance to detect such an effect at a significance level of α=0.05 yielded a sample size of 28 participants (estimation performed using G*Power (Faul et al., 2007 version 3.1) for a repeated-measures ANOVA with a three-level within-subject factor). “

Further, we did not aim to claim that endogenous opioids do not affect the perception of pain relief. Our phrasing in describing the results was in several instances too bold. The aim of the pharmacological manipulations was to investigate effects of dopamine and endogenous opioids on endogenous modulation of perceived intensity of pain relief. Here, we expected dopamine to enhance such endogenous modulation and naltrexone to reduce this modulation. The higher average pain modulation under naltrexone compared to placebo found in VAS ratings (naltrexone: -10.09, placebo: -7.31, see Table 1) suggests an increase in pain modulation by naltrexone compared to placebo, although this did not reach statistical significance, which is the opposite of what we had expected (see comment #11). Therefore, we concluded that we have no evidence to support our hypothesis of reduced endogenous modulation of pain relief by naltrexone. We do not want to claim that there are no effects of endogenous opioids on pain modulation. Although Bayesian statistics might be used to support such an interpretation, we think this might be misleading in our context here due to the considerations on the lack of power (which also affects null-hypothesis testing in Bayesian statistics) and the lack of a direct manipulation check mentioned above. Since we expected opposite effects of levodopa and naltrexone on pain modulation, we did not intend to compare these effects directly to avoid a distortion of the results. According to our hypotheses, we expected to see increased modulation of pain relief with enhanced dopamine availability and decreased modulation of pain relief with blocking of opioid receptors (see also comment #11). However, we had no a priori assumptions on potential differences in the absolute changes induced by the drug manipulations. Based on these considerations, we did now not include further direct comparisons of the effects of both drugs. Rather, we carefully went through the manuscript to tone down the descriptions and interpretations of our null findings and adjusted the respective section of the discussion to better reflect this interpretation.

3) The authors run a large number of tests. They test a number of different – and very interesting – hypotheses so this is not unreasonable, but a bit more information on error control would be useful. Note that I do not recommend an overall Bonferroni or related approach, but perhaps family-wise correction for each hypothesis test would be appropriate.

We thank the reviewer for pointing out that precise information on error control was missing. As pointed out by the reviewer, most tests reported in the manuscript relate to different hypotheses. As such, we first tested whether the manipulation of controllability within the wheel of fortune game affected perceived pain within each drug condition using mixed models. Here, the post-hoc tests used to compare the active and the passive condition (as reported in Figure 2) were not corrected for multiple comparisons, for the simple reason that there were no multiple comparisons. However, we agree that for these analyses it would be also reasonable to control for multiple testing across all drug conditions, resulting in familywise error correction for three comparisons. Accordingly, we added Bonferroni-Holm corrections to the post-hoc tests across the drug conditions. We added a more detailed explanation of this procedure in the Materials and methods section (page 33, lines 889ff):

“Here, the interaction effect indicates that active controllability in test trials has different effects for pain relief and pain increase. We used post-hoc tests to confirm that pain intensity was lower in the test compared to the control condition in win trials and equal or higher in the test compared to control condition in lose trials. We used the Bonferroni-Holm method for family-wise error correction of these post-hoc tests across drug conditions.”

In a next step, we tested whether pain modulation due to winning or losing in the wheel of fortune game differed between the drug conditions. Here, post-hoc tests for comparing pain modulation between drugs (as reported in Figure 3) were corrected for multiple comparisons using Tukey adjustment, the standard method for post-hoc pairwise comparisons of means. The same method was applied for comparing proportions of choices in favor of the option associated with a higher chance to win pain relief between drugs as well as for comparing effects of prediction errors (PE) on pain modulation between drugs (specifically for the posthoc comparisons of the linear trend of the predictor PE). The description of the use of Tukey adjustments to correct for multiple comparisons in the post-hoc tests was included in the Materials and methods section (page 35, line 954f).

4) The effects were found in one (pain intensity ratings) but not the other (behaviorally assessed pain perception) outcome measure. This weakens the findings and should at least be critically discussed.

We thank the reviewers for highlighting this important aspect. We have considered the two outcome measures as indicative of two different aspects or dimensions of the pain experience, based also on previous results in the literature. Within our procedure, the ratings indicate the momentary perception of the stimulus intensity after phasic changes in nociceptive input (outcomes), while the behavioral measure indicates perceptual within-trial sensitization or habituation in response to the tonic stimulation within each trial. Supporting the assumption of such two different aspects, it has been shown before that pain intensity ratings and behavioral discrimination measures can dissociate (Hölzl et al., 2005). In line with the assumption that both outcome measures assess different aspects of the pain experience, a differential effect of controllability on these two outcome measures is conceivable. Similarly, Becker et al. (2015), using a very similar experimental paradigm, did only find endogenous pain facilitation in the losing condition of the wheel of fortune game in pain ratings but not in the behavioral outcome measure, while they found endogenous inhibition in both measures. Compared to Becker et al. (2015), we implemented here smaller changes in stimulation intensity as outcomes in the wheel of fortune game (-3°C vs -7°C for win trials, +1°C vs +5°C for lose trials), potentially resulting in the differential effects here. Nevertheless, we agree that this reasoning needs a more explicit discussion in the manuscript and we included the following sentences to the Discussion section (page 22, lines 541ff):

“Although we did not assess the affective component of the relief experience, we implemented two outcome measures that are assumed to capture independent aspects of the pain experience: VAS ratings indicate perception of phasic changes (outcomes), while the behavioral measure indicates perceptual within-trial sensitization or habituation in response to the tonic stimulation within each trial. We found enhanced endogenous modulation by controllability and unpredictability in the VAS ratings, in line with the view that endogenous modulation enhances behaviorally relevant information. In contrast, the within-trial sensitization did not differ between the active and passive conditions under placebo. In contrast, in a previous study using a similar experimental paradigm Becker et al. (2015) found a reduction of within-trial sensitization after pain relief outcomes by controllability. Compared to this study, we implemented here smaller changes in stimulation intensity as outcomes in the wheel of fortune (-3 °C vs -7 °C for pain relief), potentially explaining the differential results.“

5) The instructions given to the participants should be specified. Moreover, it is essential to demonstrate that the instructions do not yield differences in other factors than controllability (e.g., arousal, distraction) between test and control trials. Otherwise, the main interpretation of a controllability effect is substantially weakened.

Thanks for pointing out that specific information on instructions given to the participants was missing. We agree that factors other than controllability would confound the interpretation of differences between test and control trials. We aimed minimizing nonspecific effects of arousal and/or distraction while still giving all needed information with our instructions (see below). In addition, control and test trials were kept as similar as possible. In order to check for unspecific effects of arousal and/or distraction, we also included lose trials in the game as an additional control condition. For clarifying participants’ instructions, we added the following paragraph to the Materials and methods section (page 30, lines 767ff): “The participants were instructed that there were two types of trials: trials in which they could choose a color to bet on the outcome of the wheel of fortune and trials in which they had no choice. Specifically, they were told that in the first type of trials they could use the left and right mouse button, respectively, to choose between the pink and blue section of the wheel of fortune. Participants were further instructed that if the wheel lands on the color they had chosen they will win, i.e. that the stimulation temperature will decrease, while if the wheel lands on the other color, they will lose, i.e. that the stimulation temperature will increase. For the second type of trials, participants were instructed that they could not choose a color, but were to press a black button, and that after the wheel stopped spinning the temperature would by chance either increase, decrease, or remain constant.”

In general, both arousal and distraction can be assumed to affect pain perception. If the active condition in the wheel of fortune resulted in higher arousal and/or distraction this should result in comparable effects on intensity ratings in both the win and lose outcomes compared to the passive condition. In contrast, controllability is expected to have opposite effects on pain perception in win and lose trials (decreased pain perception after winning and increased pain perception after losing in the active compared to the passive condition). These opposite effects of controllability are tested by the interaction ‘outcome × trial type’ when fitting separate models for each drug condition, which should be zero if unspecific effects of arousal and/or distraction predominated. Instead, we found a significant interaction in these models, confirming opposing effects of controllability in win and lose outcomes and contradicting such unspecific effects. We added this reasoning, to the Results section (page 5, lines 108ff) to better highlight this line of reasoning, as follows:

“To test whether playing the wheel of fortune induced endogenous pain inhibition by gaining pain relief during active (controllable) decision-making, a test condition in which participants actively engaged in the game and ‘won’ relief of a tonic thermal pain stimulus in the game was compared to a control condition with passive receipt of the same outcomes (Figure 1). As a further comparator the game included an opposite (‘lose’) condition in which participants received increases of the thermal stimulation as punishment. This active loss condition was also matched by a passive condition involving receipt of the same course of nociceptive input. Comparing the effects of active versus passive trials between the pain relief and the pain increase condition (interaction ‘outcome × trial type’) allowed us to test for unspecific effects such as arousal and/or distraction. If effects seen in the active compared to the passive condition were due to such unspecific effects, then actively engaging in the game should affect comparably pain in both win and lose trials. In contrast, if the effects were due to increased controllability, pain inhibition should occur in win trials and pain facilitation in lose trials.”

6) The blinding assessment does not rule out that the volunteers perceived the difference between placebo on the one hand and levodopa/naltrexone on the other hand. It is essential to directly show that the participants were not aware of this difference.

We based our assessment of blinding on the fact that for none of the drug conditions the frequency of guessing correctly which drug was ingested was above chance (see Results section, page 8, lines 201ff). In addition, the frequency of side effects reported by the participants did not differ between the three drug conditions, supporting this notion indirectly. However, we agree with the reviewer that this does not rule out completely that participants may have perceived a difference between the placebo and the levodopa/naltrexone conditions. We ran additional analyses to test whether participants were more likely to answer correctly that they had ingested an active drug and whether they were more likely to report side effects in the active drug conditions compared to the placebo condition. In 7 out of 28 placebo sessions (25%) the participants assumed incorrectly to have ingested one of the active drugs. In 12 out of 43 drug sessions (21.8%) the participants assumed correctly that they had ingested one of the active drugs. These frequencies did not differ between placebo sessions on the one hand and the levodopa and naltrexone active drug sessions on the other hand (χ(1) = 0.11, p = 0.737). In 9 out of 28 placebo sessions (32.1%) and in 23 out of 55 drug sessions (41.8%) participants reported to be tired at the end of the session. The frequency of reporting tiredness did not significantly differ between placebo sessions on the one hand and drug sessions on the other hand (χ(1) = 1.06, p = 0.304). No other side effects were reported. We added the following information to the Results section (page 8, lines 203ff):

“In 32 out of 83 experimental sessions subjects reported tiredness at the end of the session. However, the frequency did not significantly differ between the three drug conditions (χ(2) = 2.17, *p* = 0.337) or between the placebo condition compared to the levodopa and naltrexone condition (χ(1) = 1.06, *p* = 0.304). No other side effects were reported. To ensure that participants were kept blinded throughout the testing, they were asked to report at the end of each testing session whether they thought they received levodopa, naltrexone, placebo, or did not know. In 43 out of 83 sessions that were included in the analysis (52%), participants reported that they did not know which drug they received. In 12 out of 28 sessions (43%), participants were correct in assuming that they had ingested the placebo, in 6 out of 27 sessions (22%) levodopa, and in 2 out of 28 sessions (7%) naltrexone. The amount of correct assumptions differed between the drug conditions (χ(2) = 7.70, *p* = 0.021). However, posthoc tests revealed that neither in the levodopa nor in the naltrexone condition participants guessed the correct pharmacological manipulation significantly above chance level (*p’s* > 0.997) and the amount of correct assumptions did not differ significantly between placebo compared to levodopa and naltrexone sessions (χ(1) = 0.11, *p* = 0.737), suggesting that the blinding was successful.”

7) The effects of novelty seeking have been assessed in the placebo and the levodopa but not in the naltrexone conditions. This should be explained. Assessing novelty seeking effects also in the naltrexone condition might represent a helpful control condition supporting the specificity of the effects in the naltrexone condition.

We thank the reviewer for this interesting suggestion. Indeed, we did not report the association of pain modulation with novelty seeking in the naltrexone condition, because we did not have an a-priori hypothesis for this relationship. We now included correlations for all three drug conditions, testing if higher novelty seeking was associated with greater perceptual modulation in the active vs. passive condition. In line with comment 3, we applied a correction for multiple comparisons here (Bonferroni-Holm correction). This correction caused the correlation in the placebo condition to be no longer significant with an adjusted p-value of 0.073 (r = -0.412), while the correlation stays significant in the levodopa condition (r = -0.551, p = 0.013). Because of a reasonable effect size of the correlation under placebo (i.e. r = -0.412), we still report this correlation to highlight the increase under levodopa, while emphasizing that this correlation not significant We carefully toned down the interpretation of this correlation to reflected the change in significance with the correction for multiple testing.

We added the following information in the Results section (page 16, lines 389ff):

“Previous data suggest that endogenous pain inhibition induced by actively winning pain relief is associated with a novelty seeking personality trait: greater individual novelty seeking is associated with greater relief perception (pain inhibition) induced by winning pain relief (Becker et al., 2015). Similar to these results, we found here that endogenous pain modulation, assessed using self-reported pain intensity, induced by winning was associated with participants’ scores on novelty seeking in the NISS questionnaire (Need Inventory of Sensation Seeking; Roth and Hammelstein, 2012; subscale ‘need for stimulation’ (NS)), although this correlation failed to reach statistical significance after correction for multiple comparisons using Bonferroni-Holm method (*r* = -0.412, *p* = 0.073). A significant association between novelty seeking and endogenous pain modulation was found in the levodopa condition (*r* = 0.551, *p* = 0.013). More importantly, the higher a participants’ novelty seeking score in the NISS questionnaire, the greater the levodopa-related endogenous pain modulation when winning compared to placebo (NISS NS: *r* = -0.483, *p* = 0.034 Figure 7). In contrast, higher novelty seeking scores were not correlated with stronger pain modulation induced by winning in the naltrexone condition (*r* = 0.153, *p* = 0.381) and the naltrexone induced change in pain modulation showed no significant association with novelty seeking (*r* = 0.239, *p* = 0.499). Pain modulation after losing was not associated with novelty seeking in placebo (*r* = 0.083, *p* = 0.866), levodopa (*r* = -0.164, *p* = 0.783), or naltrexone (*r* = 0.405, *p* = 0.133).

No significant correlations with NISS novelty seeking score were found for behaviorally assessed pain modulation in the placebo, levodopa and naltrexone conditions during pain relief or pain increase (|*r*|*’s* < 0.35, *p’s* > 0.238). Similarly, the difference in pain modulation during pain relief or pain increase between the levodopa and the placebo condition and between the naltrexone and the placebo condition did also not correlate with novelty seeking (|*r*|’s < 0.22, *p’s* > 0.576).”

We also edited the interpretation of the correlation in the Discussion (page 20, lines 575ff):

“Overall, all three predictions were largely borne out by the data: relief perception as measured by VAS ratings was enhanced by controllability, unpredictability and showed a medium sized – although not significant – association with the individual novelty-seeking tendency,”

8) The writing of the manuscript is sometimes difficult to follow and should be simplified for a general readership. Sections on the information-processing account of endogenous modulation in the introduction (lines 78-93), unpredictability and endogenous pain modulation in the results (lines 278-331) are quite extensive and add comparatively little to the main findings. These sections might be shortened and simplified substantially. Moreover, providing a clearer structure for the discussion by adding subheadings might be helpful.

We have reworked the manuscript to make it easier to follow. Specifically, we reworked the Introduction section to simplify it and to make it more concise. Further, we also shortened the extensive descriptions of modeling procedures that are not central for understanding the main findings. However, based on our responses to the comments #5, #9, #10, #12, and #13, here, we added some information. We think that these additions make it easier to follow the manuscript and our line of arguments, and to understand the applied analysis strategies.

9) Effect sizes are generally small. This should be acknowledged and critically discussed. Moreover, effect sizes are given in the figures but not in the text. They should be included to the text or at least explicitly referred to in the text.

We agree that the effect sizes we report appear generally small. Importantly, the effect sizes were calculated by dividing differences in marginal means by the pooled standard deviation of the residuals and the random effects to obtain an estimate of the effect size of the underlying population rather than only for our sample. This procedure was used for the purpose of achieving more generalizable estimates. Due to considerable variance between subjects in our sample, this procedure resulted in comparatively small effect sizes. Nevertheless, we think this calculation of effects sizes results in more informative values because they can be viewed as estimates of population effects. We added specific information on the calculation of the effect sizes and a brief explanation that this procedure results in comparatively small effect sizes estimates to the Materials and methods and to the Results section (see below). In addition, we included standardized effect sizes whenever we report the respective post-hoc comparisons in the Results section.

“Effects sizes were calculated by dividing the difference in marginal means by the pooled standard deviation of the random effects and the residuals providing an estimate for the underlying population (Hedges, 2007).” (Materials and methods section, page 35, lines 951ff)

“We used post-hoc comparisons to test direction and significance of differences in either outcome condition and report standardized effect sizes for these differences. Note that all reported effect sizes account for random variation within the sample, providing an estimate for the underlying population; due to considerable variance between participants in the present study, this results in comparatively small effect sizes.” (Results section, page 5, lines 129ff)

10) Given that the Results section comes before the Methods section, it would be helpful to include some method and experimental design-related information crucial for the understanding of the results in the Results section. For example, how long was the thermal stimulus? What was the baseline temperature? etc. Maybe this information can be included in the caption of Figure 1.

We thank the reviewer for this helpful suggestion. We agree that due to the order of the manuscript sections, more information on experimental design and the statistical analysis strategies should be included in the Results section. Accordingly, we included more detailed information on the analysis strategies in the Results section (please see responses to comments #5 and #9). In addition, we added more detailed information on the experimental design and information such as the duration of the stimuli and the baseline temperature to the caption of Figure 1 (Results section, page 6, lines 136ff).

“Figure 1: Time line of one trial with active decision-making (test trials) of the wheel of fortune game. Experimental pain was implemented using contact heat stimulation on capsaicin sensitized skin on the forearm. In each trial, the temperature increased from a baseline of 30 °C to a predetermined stimulation intensity perceived as moderately painful. In each testing session, one of the two colors (pink and blue) of the wheel was associated with a higher chance to win pain relief (counterbalanced across subjects and drug conditions). Pain relief (win) as outcome of the wheel of fortune game (depicted in green) and pain increase (loss; depicted in red) were implemented as phasic changes in stimulation intensity offsetting from the tonic painful stimulation. Based on a probabilistic reward schedule for theses outcomes, participants could learn which color was associated with a better chance to win pain relief. In passive control trials and neutral trials participants did not play the game, but had to press a black button after which the wheel started spinning and landed on a random position with no pointer on the wheel. Trials with active decision-making were matched by passive control trials without decision making but the same nociceptive input (control trials), resulting in the same number of pain increase and pain decrease trials as in the active condition. In neutral trials the temperature did not change during the outcome interval of the wheel. Two outcome measures were implemented in all trial types: (i) after the phasic changes during the outcome phase participants rated the perceived momentary intensity of the stimulation on a visual analogue scale (‘VAS intensity’); (ii) after this rating, participants had to adjust the temperature to match the sensation they had memorized at the beginning of the trial, i.e. the initial perception of the tonic stimulation intensity (‘self-adjustment of temperature’). This perceptual discrimination task served as a behavioral assessment of pain sensitization and habituation across the course of one trial. One trial lasted approximately 30 s, phasic offsets occurred after approximately 10 s of tonic pain stimulation. Adapted from Becker et al. (2015).”

11) It would be helpful if the authors could provide their prior hypotheses on the drug effects. It could be a little bit confusing that the goal of using these drugs given that levodopa is a precursor of dopamine, whereas naltrexone is the opioid antagonist, i.e., the effects on the target neurotransmitters seem the opposite. Then, I wondered if the authors expected to see the opposite effects, e.g., levodopa enhances pain relief, while naltrexone inhibits pain relief, or to see similar effects, e.g., both enhance pain relief. Clarifying which direction of expected effects would be helpful for novice readers.

We thank the reviewer for pointing out that information on the expected drug effects should be explained in more detail. Indeed, we expected opposite effects of levodopa and naltrexone with respect to the effect of controllability on pain relief. Levodopa, as a precursor of dopamine, enhances dopamine availability and thus, phasic release of dopamine in response to events, for example, the reception of reward. Accordingly, we hypothesized that endogenous modulation by relief outcomes are increased in the active (reward) compared to the passive condition. In contrast, naltrexone blocks opioid receptors and as such it has been reported that naltrexone blocks placebo analgesia as a type of endogenous pain inhibition. Correspondingly, we hypothesized that naltrexone decreases endogenous pain modulation induced by actively winning pain relief compared to the passive condition. We expanded the explanation of these hypotheses in the Introduction section (page 4, lines 83ff) as follows:

“We expected increased dopamine availability to enhance phasic release of dopamine in response to rewards, and hence, to increase the effect of active compared to passive reception of pain relief. In contrast, we expected the inhibition of endogenous opioid signaling to decrease the effect of active controllability on pain relief. The latter is based on the observation that blocking of opioid receptors attenuates other types of endogenous pain inhibition such as placebo analgesia (Benedetti, 1996; Eippert et al., 2009) or conditioned pain modulation (King et al., 2013). “

12) On the "Behaviorally assessed pain perception" results in Figures 2D-F, I wonder why the results for the "pain increase" were still positive. Were the y values on the plots the temperature that participants adjusted (i.e., against the temperature right before the temperature adjustment)? or are the values showing the differences from the baseline (i.e., against the baseline temperature)?

The behavioral measure was calculated as the difference in temperatures between the memorization interval at the beginning of the trial (i.e. the predetermined temperature perceived as moderately painful) minus the self-adjusted temperature at the end of the trial so that positive values indicate sensitization (i.e. an increase in sensitivity) and negative values indicate habituation (i.e. a decrease in sensitivity) across the stimulation within on trial (i.e. approx. 30 seconds of stimulation). In general, for a stimulation of approximately 30 seconds with intensities perceived as painful, perceptual sensitization is expected to occur (Kleinböhl et al., 1999).

The outcome of the wheel of fortune game, i.e. the phasic decrease (winning) or increase (losing) in stimulation intensity, should indeed have opposite effects on this sensitization. A decrease in nociceptive input negatively reinforces pain perception, as seen in stronger sensitization in win trials, while an increase in nociceptive input punishes pain perception, as seen in reduced perceptual sensitization in lose trials. Using the a very similar task, Becker et al. (2015) found values indicating habituation within trials with temperature increases in lose outcomes. However, in this previous study, increases of +5°C were used for lose outcomes (as compared to +1 °C in the present study). Thus, in the present study the comparatively small increase in absolute stimulation temperature may not have been sufficient to induce within trial habituation to the tonic heat pain stimulation.

Nevertheless, independent of the effect of the outcome (increase or decrease of the stimulation intensity) our focus was on the additional effect that controllability (active vs. passive condition) had on the perception of the underlying tonic stimulation within each outcome condition (i.e. on the same nociceptive input). Here we expected to see endogenous inhibition after winning and endogenous facilitation after losing in the active compared to the passive condition.

We added more detailed information on the calculation of the behavioral measure and the expected perceptual modulation within each trial due to the stimulus duration in the Methods section as well as in the Results section.

Methods section (page 30, lines 779ff):

“After this rating, participants had to adjust the stimulation temperature themselves to match the temperature they had memorized at the beginning of the trial. This self-adjustment operationalizes a behavioral assessment of perceptual sensitization and habituation within one trial (Becker et al., 2011, 2015; Kleinböhl et al., 1999). Participants adjusted the temperature using the left and right button of the mouse to increase and decrease the stimulation temperature. The behavioral measure was calculated as the difference in temperatures in the memorization interval at the beginning of each trial minus this self-adjusted temperature at the end of each trial. Positive values, i.e. self-adjusted temperatures lower than the stimulation intensity at the beginning of the trial, indicate perceptual sensitization, while negative values indicate habituation.” Results section (page 7, lines 177ff):

“Positive values (i.e. lower self-adjusted temperatures compared to the stimulation intensity at the beginning of the trial) indicate perceptual sensitization across the course of one trial of the game, negative values indicate habituation. For tonic stimulation at intensities that are perceived as painful, perceptual sensitization is expected to occur (Kleinböhl et al., 1999). Differences between the outcome conditions (win, lose) reflect the effect of the phasic changes on the perception of the underlying tonic stimulus. Differences between active and passive trials reflect the effect of controllability on this perceptual sensitization within each outcome condition.”

13) Some more methods information should be included in the Results section for ease of reading, e.g. which statistical model is used to produce each result, whether session order was included etc.

We thank the reviewer for this suggestion. We added more information, particularly on the statistical analyses, in the Results section, also in line with our responses to comments #5, #9, #10, and #12. In addition, we added information on the statistical models in the Results section:

Page 6, lines 116ff:

“We implemented two outcome measures, an explicit rating of perceived intensity after pain relief or increase, and a behavioral measure of perceptual sensitization or habituation to the underlying tonic stimulation within each trial of the game (see Figure 1). The effects of controllability on pain perception were tested in separate linear mixed effects models predicting the outcome measures by the outcome condition, the trial type (active test trials vs. passive control trials), and their interaction in each drug condition.” Page 10 lines 222ff:

“Effects of pharmacological manipulations on endogenous modulation by active controllability We next examined whether endogenous modulation of pain perception within the wheel of fortune game was affected by a levodopa and naltrexone. In addition to the models testing effects of controllability on pain perception described above, we calculated pain modulation in test trials for both outcome measures as the difference of each test trial to the mean of control trials for the respective outcome condition for each participant. We fitted linear mixed effects models to predict pain modulation by drug condition, outcome, and their interaction. As an additional covariate of no interest we included session order (and respective interaction effects) in these models, as the temporal order of sessions (independent of the order of the application of the drugs) was found to have a differential effect on win and lose outcomes (see Figure 3, figure supplement 1).”

Reviewer #1 (Recommendations for the authors):– Line 598. How were continuous pain ratings performed?

We thank the reviewer for pointing out that this information was missing and apologize for this inaccuracy. Participants performed the continuous ratings by controlling the marker on the VAS using the buttons of a computer mouse that was used as the response unit. Participants moved the slider to the left or right by pressing the left or right mouse button. The slider moved in the respective direction until the participants released the button. By this, participants were able to continuously adjust their ratings according to their perception as long as the VAS was displayed on the screen. We added the following explanation in the Materials and methods section (page 28, lines 712ff):

“The VAS scale was presented on the screen with a red marker that could be moved along the scale. Participants adjusted the position of the marker by pressing the right or left mouse button. The marker moved into the respective direction until participants released the button. The participants were instructed to adjust their rating whenever a change in their perception occured. Participants could adjust their ratings as long as the VAS scale was presented on the screen.”